# Divergent receptor proteins confer responses to different karrikins in two ephemeral weeds

Yueming Kelly Sun[1,2], Jiaren Yao[1], Adrian Scaffidi[1], Kim T. Melville [1], Sabrina F. Davies[1], Charles S. Bond [1], Steven M. Smith[3,4], Gavin R. Flematti [1] & Mark T. Waters [1,2]✉

Wildfires can encourage the establishment of invasive plants by releasing potent germination stimulants, such as karrikins. Seed germination of *Brassica tournefortii*, a noxious weed of Mediterranean climates, is strongly stimulated by $KAR_1$, the archetypal karrikin produced from burning vegetation. In contrast, the closely-related yet non-fire-associated ephemeral *Arabidopsis thaliana* is unusual because it responds preferentially to $KAR_2$. The α/β-hydrolase KARRIKIN INSENSITIVE 2 (KAI2) is the putative karrikin receptor identified in *Arabidopsis*. Here we show that *B. tournefortii* expresses three *KAI2* homologues, and the most highly-expressed homologue is sufficient to confer enhanced responses to $KAR_1$ relative to $KAR_2$ when expressed in *Arabidopsis*. We identify two amino acid residues near the KAI2 active site that explain the ligand selectivity, and show that this combination has arisen independently multiple times within dicots. Our results suggest that duplication and diversification of KAI2 proteins could confer differential responses to chemical cues produced by environmental disturbance, including fire.

[1] School of Molecular Sciences, The University of Western Australia, 35 Stirling Hwy, Perth, WA 6009, Australia. [2] Australian Research Council Centre of Excellence in Plant Energy Biology, The University of Western Australia, 35 Stirling Hwy, Perth, WA 6009, Australia. [3] School of Natural Sciences, The University of Tasmania, Hobart, TAS 7000, Australia. [4] Institute of Genetics and Developmental Biology, Chinese Academy of Sciences, 1 West Beichen Road, Beijing 100101, PR China. ✉email: mark.waters@uwa.edu.au

Environmental disturbance promotes the establishment of invasive species, posing a potent threat to global biodiversity. Changing wildfire regimes, such as increasing frequency of fires, is one of the most-relevant disturbance factors contributing to elevated invasion threat[1]. Wildfires create germination opportunities in part by releasing seed germination stimulants, such as karrikins, from burning vegetation[2,3]. Karrikins comprise a family of butenolides with six known members. In samples of smoke water generated by burning grass straw, KAR$_1$ is the most abundant karrikin, whereas KAR$_2$ is six times less abundant[4], although these proportions may differ depending on source material, preparation method, age and storage conditions[5]. These two analogues differ only by the presence of a methyl group on the butenolide ring in KAR$_1$, which is absent in KAR$_2$ (Supplementary Fig. 1a). Invasive plant species that are responsive to karrikins could utilise natural and human-induced fires to facilitate their establishment[6,7].

Karrikins can overcome seed dormancy and promote seed germination in a number of smoke-responsive species, as well as others that are not associated with fire regimes[8–10]. Furthermore, karrikins also influence light-dependent seedling development. Karrikins enhance the sensitivity of *Arabidopsis thaliana* seedlings to light by inhibiting hypocotyl elongation, stimulating cotyledon expansion and promoting chlorophyll accumulation in a dose-dependent manner[11]. Karrikins have been reported to enhance seedling survival and biomass in species such as tomato, rice and maize[12,13]. Such growth-promoting effects of karrikins, especially at critical early stages of the life cycle, have the potential to further encourage the establishment of invasive species after fire events.

*Brassica tournefortii* (Brassicaceae; Sahara mustard) is native to northern Africa and the Middle East, but is an invasive weed that blights many ecosystems with a Mediterranean climate and chaparral-type vegetation that are prone to wildfires in North America, Australia and South Africa. *B. tournefortii* seeds can persist in the soil for many seasons, undergoing wet–dry cycling that can influence dormancy and contribute to boom-bust cycles that outcompete native species[9,14]. *B. tournefortii* plants may radically alter fire frequency and intensity by influencing fuel profiles[15,16], further exacerbating the impact of fire on susceptible native ecosystems. In addition, seeds of *B. tournefortii* are particularly responsive to smoke-derived karrikins, and show a positive germination response to KAR$_1$ in the nanomolar range[10]. Accordingly, *B. tournefortii* is particularly well positioned to invade areas disturbed by fire events[17,18].

The putative karrikin receptor KARRIKIN INSENSITIVE 2 (KAI2) was identified in *Arabidopsis*, a weedy ephemeral that originated in Eurasia but is now widely distributed throughout the northern hemisphere[19–21]. *Arabidopsis* is not known to colonise fire-prone habitats, but nevertheless seeds germinate in response to karrikins in the micromolar range[22]. Unlike most smoke-responsive species that respond more readily to KAR$_1$[8,23], *Arabidopsis* responds preferentially to KAR$_2$[22]. KAI2 is an evolutionarily ancient α/β-hydrolase and a paralogue of DWARF14 (D14), the receptor for strigolactones[24,25]. Karrikins and strigolactones are chemically similar by virtue of a butenolide moiety that is necessary for bioactivity[26,27]. KAI2 and D14 have dual functions as both enzyme and receptor, but the functional significance of the enzymatic activity remains contested[28–32]. Furthermore, the basis for ligand specificity by these two highly congruent proteins remains essentially unknown.

Orthologues of *KAI2* are ubiquitous in land plants, and are normally present as a single gene copy within an ancient and highly conserved 'eu-KAI2' clade[33]. There is growing evidence that, beyond its ability to mediate karrikin responses, KAI2 has a core ancestral role in perceiving an endogenous KAI2 ligand ('KL') that regulates seed germination, seedling development, leaf shape and cuticle development[34–36]. Since its divergence from the *Arabidopsis* lineage, the tribe Brassiceae, which includes the genus *Brassica*, underwent a whole genome triplication event 24–29 million years ago[37–39]. This process might have allowed additional KAI2 copies to gain altered ligand specificity, potentially enhancing perception of environmental signals such as karrikins from smoke. Here, we report that two out of three KAI2 homologues expressed in *B. tournefortii* show distinct preferences for different karrikins. We take advantage of the relatively recent genome triplication event in the Brassiceae to identify two amino acids that are sufficient to explain these karrikin preferences and confirm this by mutagenesis. Beyond demonstrating the potential ecological significance of diversity among KAI2 homologues, our findings also reveal active site regions critical for ligand selectivity among KAI2 receptor-enzymes that are found in all land plants.

## Results

**B. tournefortii is most sensitive to KAR$_1$.** To characterise the karrikin response of *B. tournefortii*, we performed multiple physiological and molecular assays comparing KAR$_1$ activity with that of KAR$_2$. First, germination of *B. tournefortii* seeds was consistently more responsive to KAR$_1$ than KAR$_2$ at 10 nM, 100 nM, and 1 μM (Fig. 1a and Supplementary Fig. 1b–d). Second, homologues of two karrikin-responsive transcripts, *DWARF14-LIKE2* (*BtDLK2*) and *SALT TOLERANCE HOMOLOG7* (*BtSTH7*), identified on the basis of close sequence homology to their *Arabidopsis* counterparts[11,20], were significantly more highly expressed when treated with 1 μM KAR$_1$ than with 1 μM KAR$_2$ (Fig. 1b). These observed differences in seed response are not owing to differential karrikin uptake, as both KAR$_1$ and KAR$_2$ were taken up from solution at similar rates by *B. tournefortii* seeds during imbibition, as was also true for *Arabidopsis* seeds (Supplementary Fig. 1e–f). Besides promoting germination of primary-dormant seeds, karrikins also inhibited hypocotyl elongation in *B. tournefortii* seedlings, as is the case in *Arabidopsis*; again, KAR$_1$ showed a stronger effect than KAR$_2$ (Fig. 1c, d). Levels of *BtDLK2* transcripts in seedlings were also more responsive to KAR$_1$ than KAR$_2$ at a given concentration (Fig. 1e). Therefore, we conclude that *B. tournefortii* is more sensitive to KAR$_1$ than to KAR$_2$, a ligand preference that is a feature of seed germination in many karrikin-responsive species from ecosystems prone to fires[8,23].

**BtKAI2a and BtKAI2b are functional in Arabidopsis.** To establish whether there are multiple *KAI2* homologues present in *B. tournefortii*, we examined transcriptomes from seeds and seedlings. Three putative *KAI2* homologues were identified (*BtKAI2a*, *BtKAI2b* and *BtKAI2c*; Fig. 2a; Supplementary Figs. 2, 3). *BtKAI2a* grouped with *AtKAI2* and those of other Brassicaceae within a single clade, whereas *BtKAI2b* and *BtKAI2c* grouped within a clade unique to *Brassica*. This taxonomic distribution implies that *BtKAI2a* is most similar to *AtKAI2*, and that *BtKAI2b* and *BtKAI2c* are paralogues that arose via genome triplication during the evolution of *Brassica*. All three *BtKAI2* transcripts were expressed in *B. tournefortii* seeds, but only *BtKAI2a* and *BtKAI2b* could be detected in seedlings (Fig. 2b). In seeds and seedlings, *BtKAI2b* transcripts were the most abundant of the three, but there were no consistent effects on any of the transcripts upon treatment with 1 μM KAR$_1$. We also identified two *BtD14* homologues, at least one of which is functionally orthologous to *AtD14* (Supplementary Figs. 2, 4).

We performed transgenic complementation of the *Arabidopsis kai2-2* null mutant by expressing each of the three isoforms as a GFP fusion protein driven by the *Arabidopsis KAI2* promoter and 5′-UTR (*KAI2pro:GFP-BtKAI2*) to accurately reflect the native

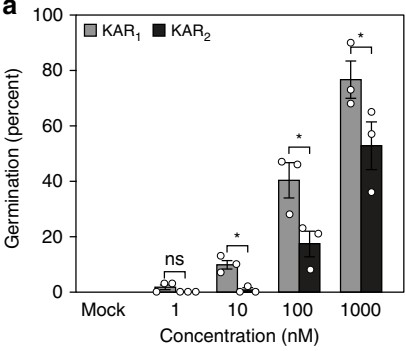

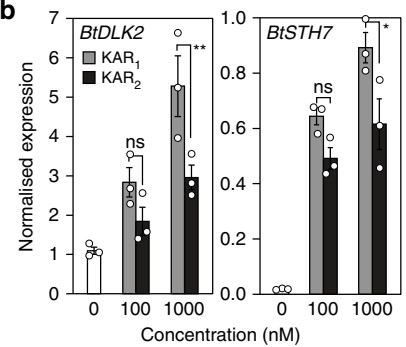

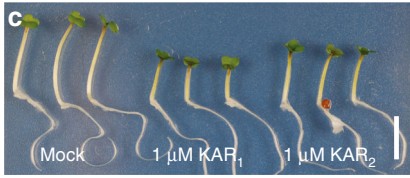

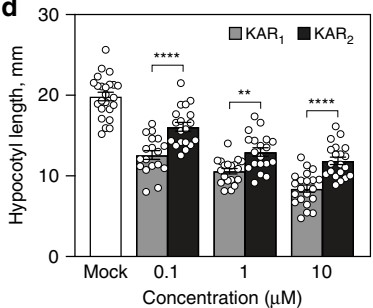

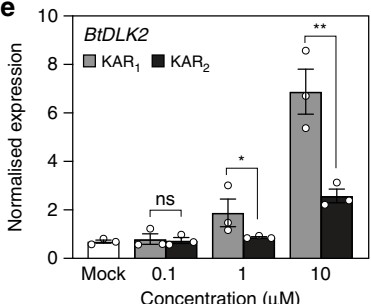

**Fig. 1 *Brassica tournefortii* is highly sensitive to KAR$_1$, the major karrikin analogue isolated from plant-derived smoke. a** Germination responses of *B. tournefortii* seed to KAR$_1$ and KAR$_2$. Data are cumulative germination after 11 days (mean ± SE; $n = 3$ biological replicates per treatment, ≥35 seeds per replicate). **b** Levels of karrikin-responsive transcripts *BtDLK2* and *BtSTH7* in *B. tournefortii* seed. Seeds were imbibed for 24 h in the dark supplemented with KAR$_1$ and KAR$_2$. Transcripts were normalised to *BtCACS* reference transcripts. Data are means ± SE, $n = 3$ biological replicates, ≥50 seeds per replicate. **c, d** Hypocotyl elongation responses of *B. tournefortii* seedlings grown for 4 days under continuous red light on water-saturated glass fibre filter paper containing 0.1% acetone (mock), KAR$_1$ or KAR$_2$. Data are means ± 95% CI of $n = 18$–24 seedlings. Scale bar: 10 mm. **e** Levels of *BtDLK2* transcripts in *B. tournefortii* seedlings grown under the same conditions as for **d**. Data are means ± SE, $n = 3$ biological replicates, ≥20 seedlings per replicate. In all panels, asterisks denote significance levels (ANOVA) between indicated conditions: *$P < 0.05$, **$P < 0.01$, ***$P < 0.001$, ****$P < 0.0001$. Source data are provided as a Source Data file.

expression profile of KAI2 in *Arabidopsis*. Such fusions of GFP with KAI2 proteins have been used previously to analyse KAI2 activity[19,40,41]. Both *BtKAI2a* and *BtKAI2b* complemented the seedling and leaf phenotypes of *kai2-2*, whereas *BtKAI2c* did not (Fig. 2c, d). GFP-BtKAI2a and GFP-BtKAI2b accumulated at consistent levels when detected using both an anti-KAI2 antibody

and an anti-GFP antibody (which negated the effects of sequence differences in the anti-KAI2 epitope region; Fig. 2e and Supplementary Fig. 3). However, we could not detect GFP-BtKAI2c protein in three independent transgenic lines with either antibody. We verified the *GFP-BtKAI2c* transcript sequences expressed in the transgenic plant material by RT-PCR and Sanger sequencing (Supplementary Fig. 5). As the *GFP-BtKAI2c* mRNA is processed faithfully in *Arabidopsis*, the apparent absence of protein is most likely a result of posttranslational events. We then tested whether BtKAI2c was generally poorly expressed in plant cells by transient overexpression in tobacco leaves using *Agrobacterium*-mediated infiltration and tobacco mosaic virus-based plant expression vectors[42]. Even with the extremely high levels of expression supported by this system, levels of myc-tagged BtKAI2c protein were substantially lower than equivalently tagged BtKAI2a and BtKAI2b (Supplementary Fig. 5). This result implies that BtKAI2c protein may be inherently unstable and/or prone to degradation in plant cells.

**BtKAI2c carries a destabilising mutation at position 98.** Numerous mutations identified by forward genetics destabilise AtKAI2 and render the protein undetectable in plant extracts[43]. Ten residues in the BtKAI2c sequence are unique to BtKAI2c within the genus *Brassica* (Supplementary Fig. 3). We predicted the likely effect of each of these residues upon protein function, relative to the *Brassica* consensus residues at the same positions, using the PROVEAN tool[44]. Among the 10 sites, R98 and E129 exceeded the cutoff value of $-2.5$ and were thus predicted to be highly deleterious to protein function, whereas D137 narrowly missed this threshold (Supplementary Table 2). Therefore, based on this predictive analysis, the native BtKAI2c protein carries at least two and possibly three substitutions that have the potential to render the protein non-functional.

R98 is potentially highly deleterious because it is situated in the active site adjacent to the catalytic serine, whereas E129 and D137 are located on the flexible hinge that links the lid and core domains (Supplementary Fig. 6a). In AtKAI2, the G133E mutation is highly destabilising[20,41], suggesting that non-conserved substitutions in the hinge region have the potential to be deleterious as well. To test the effect of these residues on BtKAI2c stability, we constructed three variants that reverted the native BtKAI2c sequence to consensus amino acids within *Brassica* at position 98 and/or in the hinge region: BtKAI2c$^{R98V}$, a quadruple mutant BtKAI2c$^{E129D;D130V;E133Q;D137E}$ (BtKAI2c$^{QUAD}$), and a quintuple mutant that combined all five substitutions (BtKAI2c$^{QUINT}$). Should these

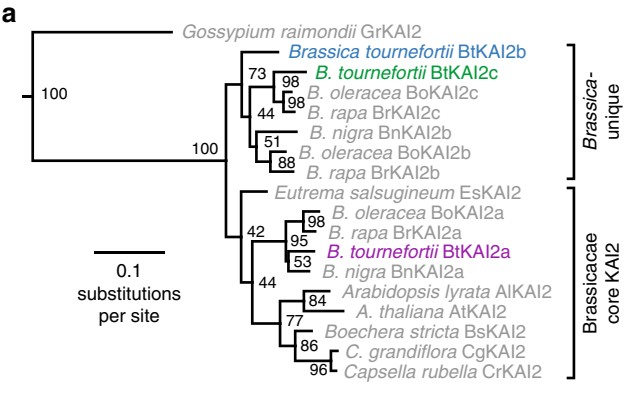

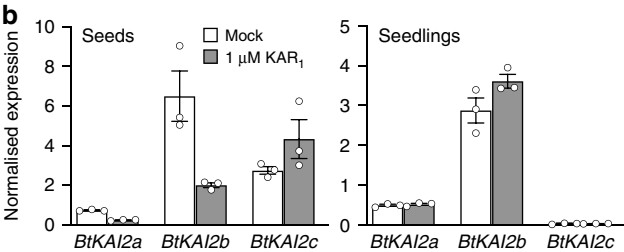

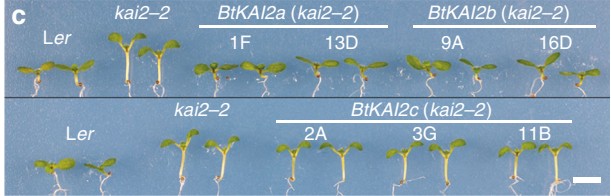

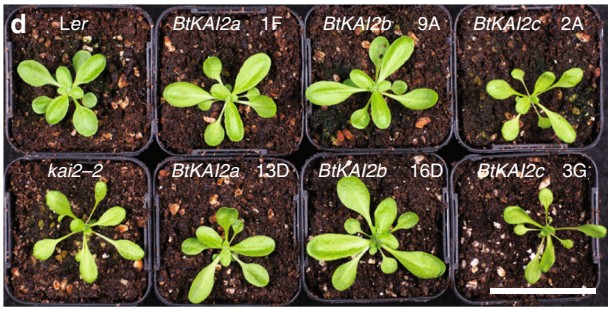

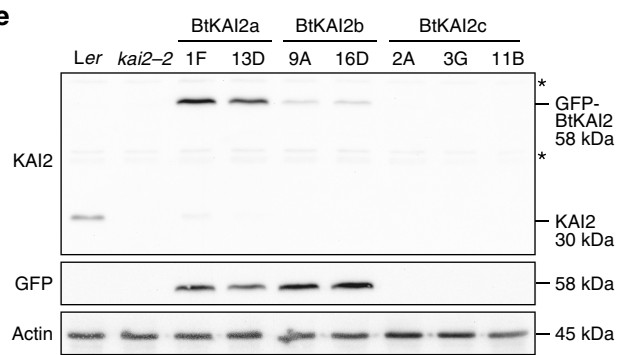

**Fig. 2 Two differentially expressed *B. tournefortii* *KAI2* homologues are functional in *Arabidopsis*. a** Maximum likelihood phylogeny of KAI2 homologues in the Brassicaceae, based on nucleotide data. Node values represent bootstrap support from 100 replicates. A eu-KAI2 sequence from *Gossypium raimondii* (Malvaceae) serves as an outgroup. Tree shown is a subset of a larger phylogeny in Supplementary Fig. 2. **b** Transcript levels of the three BtKAI2 homologues in *B. tournefortii* seeds imbibed for 24 h (left) and 4-day-old seedlings (right) treated with 0.1% acetone (mock) or with 1 μM KAR$_1$ for 24 h. **c**, **d** Seedling and rosette phenotypes of two independent transgenic lines of *Arabidopsis* homozygous for *KAI2pro:GFP-BtKAI2* transgenes. Seedlings were 7 days old and rosettes 22 days old. Scale bars: 5 mm **c**; 50 mm **d**. **e** Immunoblots of soluble proteins challenged with antibodies against KAI2 (upper panel), GFP (middle panel) or actin as a loading control (lower panel). Anti-KAI2 detects both the native AtKAI2 protein (30 kDa) and the GFP-BtKAI2 fusion proteins (58 kDa) but detects BtKAI2b relatively poorly. Non-specific bands are marked with asterisks. Protein was isolated from pools of ~50 7-day-old seedlings. Source data are provided as a Source Data file.

for stably transformed *Arabidopsis*, we found that native GFP-BtKAI2c protein was undetectable, as was the BtKAI2c$^{QUAD}$ variant (Supplementary Fig. 6b). However, BtKAI2c$^{R98V}$ and BtKAI2c$^{QUINT}$ were clearly detected. We also examined the effect of the equivalent V96R mutation on GFP-AtKAI2, and found that this mutation completely abolished expression compared with the native GFP-AtKAI2 control (Supplementary Fig. 6b). These results suggest that the R98 residue causes the loss of protein expression in BtKAI2c, and that arginine at this position is not tolerated by KAI2 proteins in general.

Early in our investigation, we had tried without success to express and purify BtKAI2c from *E. coli* for biochemical characterisation. Having restored BtKAI2c expression in plants with the R98V mutation, we next tried to express these proteins in bacteria. We found that SUMO-BtKAI2c R98V was expressed at ~10-fold higher levels than native SUMO-BtKAI2c in crude lysates, suggesting that the R98V mutation enhanced protein folding and/or stability in bacterial cells. Although it was detectable in crude lysates, SUMO-BtKAI2c was consistently intransigent to recovery following affinity chromatography; in contrast, the R98V and quintuple mutant versions were stably recovered at high levels and purity (Supplementary Fig. 6c). From these results, we conclude that native BtKAI2c is non-functional owing to protein instability induced by a highly non-conservative substitution at the active site.

**BtKAI2a and BtKAI2b show differential ligand specificity.** We further characterised the ligand specificity of BtKAI2 homologues by performing physiological and molecular assays with the stable transgenic *Arabidopsis* lines. Primary-dormant *Arabidopsis* seeds homozygous for the transgenes were tested for germination response (Fig. 3a and Supplementary Fig. 7). None of the transgenic lines completely restored germination levels to those of wild type, perhaps reflecting the need for additional, seed-specific regulatory elements not included in our transgenes, or the importance of genomic context for proper *KAI2* expression in seeds. Nevertheless, germination responses to karrikins were evident in the transgenic seeds: germination of *BtKAI2b* seeds was more responsive to KAR$_1$ than KAR$_2$ at 1 μM, whereas no significant difference was observed for *BtKAI2a* seeds. As expected, *BtKAI2c* seeds were indistinguishable from the *kai2* control and insensitive to karrikins. We also found that levels of *KAR-UP F-BOX 1* (*KUF1*) transcripts, which are responsive to karrikins in *Arabidopsis* seeds[11], were only slightly induced by karrikins in

amino acids be crucial to protein stability, we expected to restore BtKAI2c expression.

First, we tried expressing the native and mutated BtKAI2c variants in tobacco leaves using *Agrobacterium*-mediated infiltration. The vectors were based on those used to generate the *GFP-BtKAI2c Arabidopsis* transgenics, allowing us to compare expression of the same GFP fusion proteins. As was the case

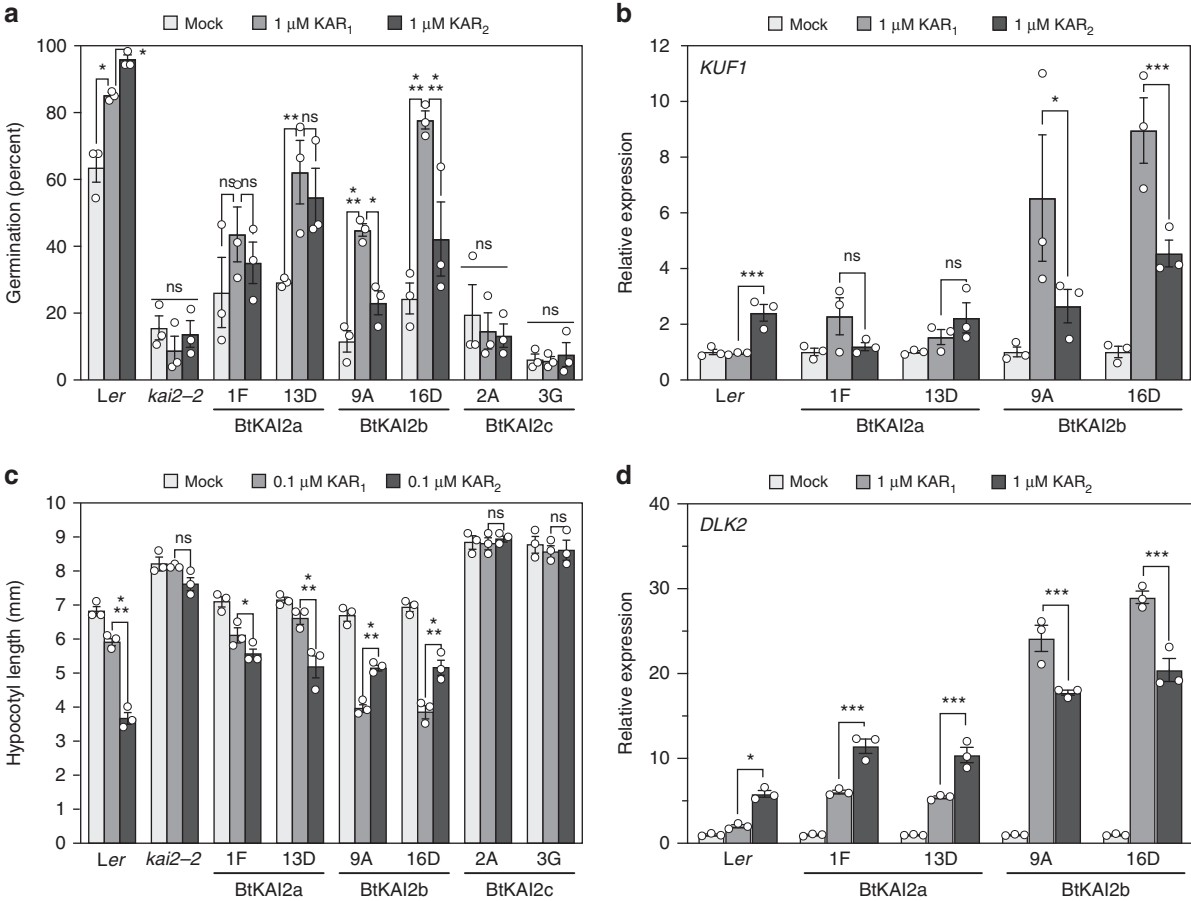

**Fig. 3 Functional divergence between BtKAI2 homologues. a** Germination responses of primary-dormant *Arabidopsis* seed homozygous for *KAI2pro:GFP-BtKAI2* transgenes in the *kai2-2* background. Germination values were determined 120 h after sowing. Extended germination curves are shown in Supplementary Fig. 7. Data are means ± SE, n = 3 independent seed batches, 75 seeds per batch. **b** Levels of *KUF1* transcripts in *KAI2pro:GFP-BtKAI2* seeds treated with 1 μM KAR$_1$ or KAR$_2$ for 96 h (see Methods). Expression was normalised to *CACS* reference transcripts and scaled to the value for mock-treated seed within each genotype. Data are means ± SE of n = 3 biological replicates. **c** Hypocotyl elongation responses of *KAI2pro:GFP-BtKAI2* seedlings treated with KAR$_1$ or KAR$_2$. Data are means ± SE of n = 3 biological replicates, 12–18 seedlings per replicate. **d** Levels of *DLK2* transcripts in 8-day-old *KAI2pro:GFP-BtKAI2* seedlings treated with KAR$_1$ or KAR$_2$ for 8 h. Expression was normalised to *CACS* reference transcripts and scaled to the value for mock-treated seedlings within each genotype. Data are means ± SE of n = 3 biological replicates. Pairwise significant differences: *$P < 0.05$, **$P < 0.01$, ***$P < 0.001$; ns, $P > 0.05$ (ANOVA). Source data are provided as a Source Data file.

*BtKAI2a* transgenic seeds, but were strongly enhanced by KAR$_1$ in *BtKAI2b* seeds (Fig. 3b).

In seedlings, both *BtKAI2a* and *BtKAI2b*, but not *BtKAI2c*, could complement the *kai2* hypocotyl elongation phenotype (Fig. 3c). Responses to karrikins in terms of hypocotyl elongation (Fig. 3c) and levels of *DLK2* transcripts (Fig. 3d) broadly agreed with the germination response with respect to karrikin preferences, with *BtKAI2b* transgenics showing a clear preference for KAR$_1$ over KAR$_2$. However, *BtKAI2a* seedlings showed a preference for KAR$_2$ that was not evident in seeds. From these experiments, we conclude that BtKAI2a is similar to AtKAI2 in terms of ligand specificity, whereas BtKAI2b has a clear preference for KAR$_1$ over KAR$_2$. As *BtKAI2b* is more highly expressed than *BtKAI2a* in *B. tournefortii* seeds and seedlings (Fig. 2b), we conclude that the ligand specificity of BtKAI2b substantially contributes to the enhanced KAR$_1$-responsiveness of this species at these stages of the life cycle.

**Positions 98 and 191 account for ligand specificity**. To investigate interactions between BtKAI2 homologues and ligands, we performed differential scanning fluorimetry (DSF) assays on purified recombinant proteins (Supplementary Fig. 8). DSF has

been used extensively for inferring the interaction of strigolactone-like compounds with D14- and KAI2-related proteins[29,41,45–48]. Racemic GR24 is a widely used synthetic strigolactone analogue that consists of two enantiomers (Supplementary Fig. 1a), of which GR24$^{ent\text{-}5DS}$ is bioactive via AtKAI2[40,49]. Catalytically inactive D14 and KAI2 variants do not respond to GR24 in DSF, suggesting that the shift in thermal response results from ligand hydrolysis and a corresponding conformational change in the receptor[29,41,46]. In DSF assays, AtKAI2 shows a specific response to >100 μM GR24$^{ent\text{-}5DS}$ but, for unclear reasons, no response to karrikins[41]. Likewise, we found that both BtKAI2a and BtKAI2b were also unresponsive to karrikins in DSF (Supplementary Fig. 9a). Therefore, we used DSF assays with GR24$^{ent\text{-}5DS}$ as a surrogate substrate as a means to analyse differences in ligand interactions between BtKAI2a and BtKAI2b. We used N-terminal 6xHIS-SUMO as a solubility tag to aid expression in *E. coli* and enhance stability. We found that the presence of the tag did not compromise the thermal destabilisation response of BtKAI2b to GR24$^{ent\text{-}5DS}$ (Supplementary Fig. 9b), and therefore, we used intact 6xHIS-SUMO fusion proteins (prefixed with 'SUMO' throughout) for all other experiments.

We found that SUMO-AtKAI2 and SUMO-BtKAI2a showed similar responses to >100 μM GR24$^{ent\text{-}5DS}$, but the response of SUMO-BtKAI2b was clear at >25 μM (Supplementary Fig. 9c). We then used a lower range of GR24$^{ent\text{-}5DS}$ concentrations (0–50 μM) to determine the threshold for response, which we defined as a statistically significant reduction in the maximal rate of change in fluorescence at the melting temperature of the protein ($T_m$). Although SUMO-BtKAI2a showed only a weak and non-significant response at 40 and 50 μM GR24$^{ent\text{-}5DS}$, SUMO-BtKAI2b responded significantly at 10 μM and above (Fig. 4a). These results suggest that BtKAI2b is more sensitive than BtKAI2a to GR24$^{ent\text{-}5DS}$, and that BtKAI2a is most like AtKAI2 in this respect.

BtKAI2a and BtKAI2b differ in primary amino-acid sequence at just 14 positions (Supplementary Fig. 3). We postulated that differences in ligand specificity might be determined by amino acids in the vicinity of the ligand-binding pocket. Protein structural homology models revealed only two residues that differ in this region: V98 and V191 in BtKAI2a, and L98 and L191 in BtKAI2b; the corresponding residues in AtKAI2 and AtD14 are valines (Supplementary Fig. 8). Residue 98 is immediately adjacent to the catalytic serine at the base of the pocket. Residue 191 is located internally on αT4 of the lid domain which, in AtD14, is associated with a major rearrangement of protein structure upon ligand binding that reduces the size of the pocket[31]. Homology modelling suggests only subtle differences in the size and shape of the primary ligand binding pocket of BtKAI2a and BtKAI2b (Supplementary Fig. 8). To determine whether these residues are pertinent to ligand specificity, we replaced the two valine residues of BtKAI2a with leucine residues, generating the variant BtKAI2a$^{L98;L191}$, and vice versa for BtKAI2b, generating the variant BtKAI2b$^{V98;V191}$ (Supplementary Fig. 8). In DSF assays, we found that exchanging the two residues was sufficient to switch the original responses, such that SUMO-BtKAI2a$^{L98;L191}$ responded sensitively to GR24$^{ent\text{-}5DS}$, but SUMO-BtKAI2b$^{V98;V191}$ did not (Fig. 4b). We also assessed the function of the stable SUMO-BtKAI2c$^{R98V}$ and SUMO-BtKAI2c$^{QUINT}$ variants by DSF against GR24$^{ent\text{-}5DS}$ ligand, and found that they were only weakly destabilised by this ligand, when compared with the highly sensitive BtKAI2b (Supplementary Fig. 6e). Overall, these results indicate that BtKAI2a and BtKAI2b have distinct response profiles, albeit to a synthetic ligand, and that residues 98 and 191 contribute to this difference.

We reasoned that the most robust and informative means to assess the effect of residues 98 and 191 upon the response to karrikins was to test their function directly in planta. We expressed BtKAI2a$^{L98;L191}$ and BtKAI2b$^{V98;V191}$ as GFP fusion proteins in the Arabidopsis kai2-2 null mutant background driven by the Arabidopsis KAI2 promoter (KAI2pro:GFP-BtKAI2a$^{L98;L191}$ and KAI2pro:GFP-BtKAI2b$^{V98;V191}$), and selected two independent homozygous transgenic lines for each, on the basis of protein expression level and segregation ratio (Supplementary Fig. 10). Using two different assays, we found that substitutions between BtKAI2a and BtKAI2b at positions 98 and 191 also reversed karrikin preference in Arabidopsis seedlings both in terms of hypocotyl elongation and DLK2 transcript levels (Fig. 4c, d; Supplementary Fig. 11). Most prominently, the clear preference of BtKAI2b for KAR$_1$ was unambiguously reversed to a preference for KAR$_2$ in BtKAI2b$^{V98;V191}$ transgenics, effectively recapitulating the response of the native BtKAI2a protein. We also examined the responses of seeds in these transgenic lines, and found that BtKAI2a$^{L98;L191}$ seeds germinated with a clear preference for KAR$_1$, whereas BtKAI2b$^{V98;V191}$ seeds showed no clear preference for either karrikin (Supplementary Fig. 12). Notably, this germination response pattern is the opposite to that observed for seeds expressing the native BtKAI2 proteins, with BtKAI2a

exhibiting no preference for either karrikin and BtKAI2b a preference for KAR$_1$ (Fig. 3a, Supplementary Fig. 7). Taken together, these results demonstrate that the residues at positions 98 and 191 largely determine differences in karrikin specificity between BtKAI2a and BtKAI2b, at least in the context of Arabidopsis seedlings.

**Co-dependency between positions 96 and 189 among angiosperms.** We wished to gain further insight into the functional importance of diversity at positions 96 and 189 of KAI2 proteins (based on AtKAI2 notation, equivalent to positions 98 and 191 in BtKAI2a and BtKAI2b). We analysed 476 distinct KAI2 homologues from 441 angiosperm species (spanning eudicots, monocots and magnoliids, but excluding B. tournefortii) using data collected from the 1000 Plant Transcriptomes project[50]. We found that at positions 96 and 189, valine was by far the most common amino acid (93.1% and 93.5% of sequences, respectively; Fig. 5a and Supplementary Data 1). L96 was observed in 6.1% of sequences, whereas L189 was even rarer (2.1%). The L96;V189 combination was observed in 14 sequences (2.9%), whereas V96; L189 was observed in only one homologue from Ternstroemia gymanthera (Pentaphylacaceae; 0.2%). Notably, this species also expressed a second KAI2 homologue with the canonical V96; V189 combination. If the frequencies of valine and leucine at positions 96 and 189 were independent, the L96;L189 combination would be expected to occur less than once among 476 sequences (0.13%). However, the coincidence of L96 and L189 was observed in nine sequences (1.9%), representing a significant 15-fold enrichment for the L96;L189 pair over that expected by chance alone ($\chi^2 = 115.6$; $P < 0.0001$).

Among the nine newly-identified KAI2$^{L96;L189}$ sequences, eight were restricted to eudicots, distributed evenly between both the superrosids (Brassicales, Fabales and Saxifragales; four species) and superasterids (Caryophyllales; four species). The ninth sequence, co-expressed with another transcript encoding KAI2$^{V96;V189}$, was identified from the basal dicot Hakea drupacea (syn. Hakea suaveolens; Proteaceae). Strikingly, this species is native to southwestern Australia, but is recognised as an invasive weed in South Africa that relies upon fire to liberate seeds from the canopy[51]. The L96;L189 combination therefore seems to have arisen independently on multiple occasions. Furthermore, within the order Saxifragales, Daphniphyllum macropodum (KAI2$^{L96;L189}$) belongs to a clade of species that all express KAI2$^{L96;V189}$ homologues (Fig. 5b). Although this clade is only represented by five sequences and four species, this distribution pattern suggests that a V96L mutation occurred first in the common ancestor of the Hamamelidaceae, Cercidiphyllaceae and Daphyniphyllaceae, followed by V189L in the lineage leading to D. macropodum. Overall, this analysis suggests that positions 96 and 189, while not in direct contact with each other in the protein tertiary structure (Supplementary Fig. 8), likely exhibit functional interdependence. Valine is most probably the ancestral state at both positions, but leucine at position 96 is also tolerated. Although rare, leucine at position 189 is generally found in combination with leucine at position 96, and tentatively, a V96L mutation might potentiate a subsequent V189L mutation.

The above analysis suggested that a V96L substitution might be sufficient to confer altered substrate specificity upon KAI2 proteins. To test this hypothesis, we generated transgenic Arabidopsis seedlings expressing GFP-BtKAI2b proteins with just a single amino-acid substitution from leucine to valine at either position 98 or 191 (i.e., KAI2pro:GFP-BtKAI2b$^{V98}$ or KAI2pro:GFP-BtKAI2b$^{V191}$), and examined their response to KAR$_1$ and KAR$_2$ in hypocotyl elongation assays. We found that the L98V substitution was sufficient to invert the karrikin

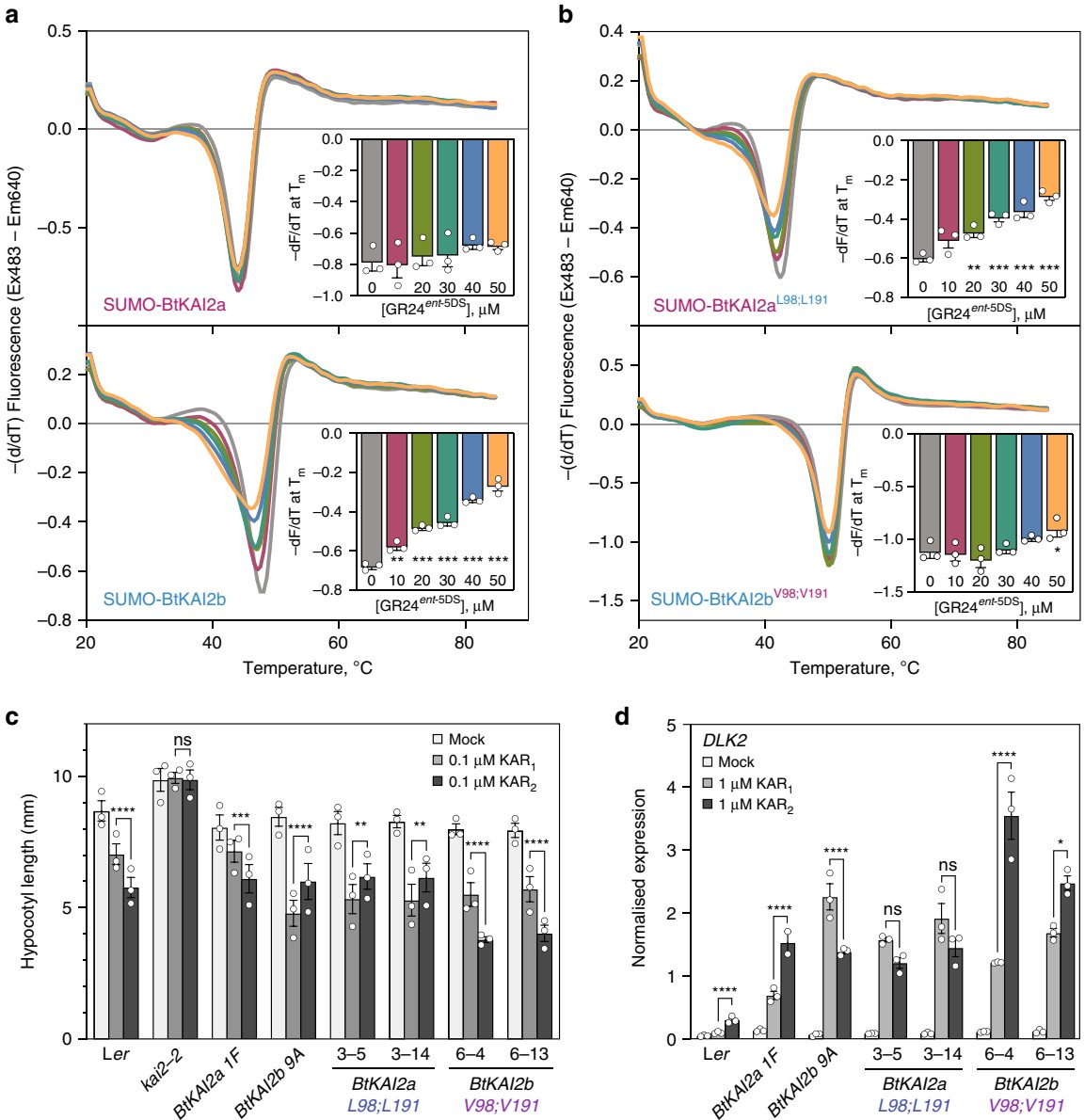

**Fig. 4 Two residues account for ligand specificity between BtKAI2a and BtKAI2b. a** DSF curves of SUMO-BtKAI2a and SUMO-BtKAI2b fusion proteins treated with 0–50 µM GR24$^{ent\text{-}5DS}$, a KAI2-bioactive ligand. Each curve is the average of three sets of reactions, each comprising four technical replicates. Insets plot the minimum value of $-(dF/dT)$ at the melting point of the protein as determined in the absence of ligand (means ± SE, $n = 3$). Significant differences from untreated control: *$P < 0.05$, **$P < 0.01$, ***$P < 0.001$ (ANOVA). **b** DSF curves of SUMO-BtKAI2a$^{L98;L191}$ and SUMO-BtKAI2b$^{V98;V191}$ fusion proteins treated with 0–50 µM GR24$^{ent\text{-}5DS}$. **c** Hypocotyl elongation responses of *Arabidopsis* expressing GFP-BtKAI2a$^{L98;L191}$ and GFP-BtKAI2b$^{V98;V191}$ fusion proteins and treated with KAR$_1$ or KAR$_2$. Data are a summary of three experimental replicates performed on separate occasions, each comprising 25–40 seedlings per genotype/treatment combination. Data for each replicate are shown in Supplementary Fig. 11. Error bars are SE, $n = 3$ experimental replicates; each dot corresponds to the mean value derived from each replicate. Asterisks denote significant differences: *$P < 0.05$, **$P < 0.01$, ***$P < 0.001$, ****$P < 0.0001$ (linear mixed model with experimental replicate as a random effect; specific pairwise comparisons using Tukey's HSD correction). **d** Levels of *DLK2* transcripts in 8-day-old seedlings of the same transgenic lines as above treated with 1 µM KAR$_1$, 1 µM KAR$_2$, or 0.1% acetone (mock) and harvested 8 h later. Expression was normalised to *CACS* reference transcripts. Data are means ± SE of $n = 3$ pools of ~50 seedlings treated in parallel. Asterisks denote significant differences as above (two-way ANOVA; specific pairwise comparisons using Tukey's HSD correction). Source data are provided as a Source Data file.

preference of GFP-BtKAI2b in three independent homozygous transgenic lines (Fig. 5c; Supplementary Fig. 13). Although this result does not rule out an additional contribution towards ligand specificity from L191, it is consistent with the higher frequency of L96 compared to L189 observed among the angiosperms, and suggests that substitution at position 96 may have been selected relatively frequently in the evolution of KAI2 function.

## Discussion

The plant α/β-hydrolases KAI2 and D14 are characterised by responsiveness towards butenolide compounds, including endogenous strigolactones and strigolactone-related compounds, abiotic karrikins derived from burnt vegetation, and synthetic strigolactone analogues with a wide array of functional groups. Direct evidence for a receptor-ligand relationship between karrikins and KAI2

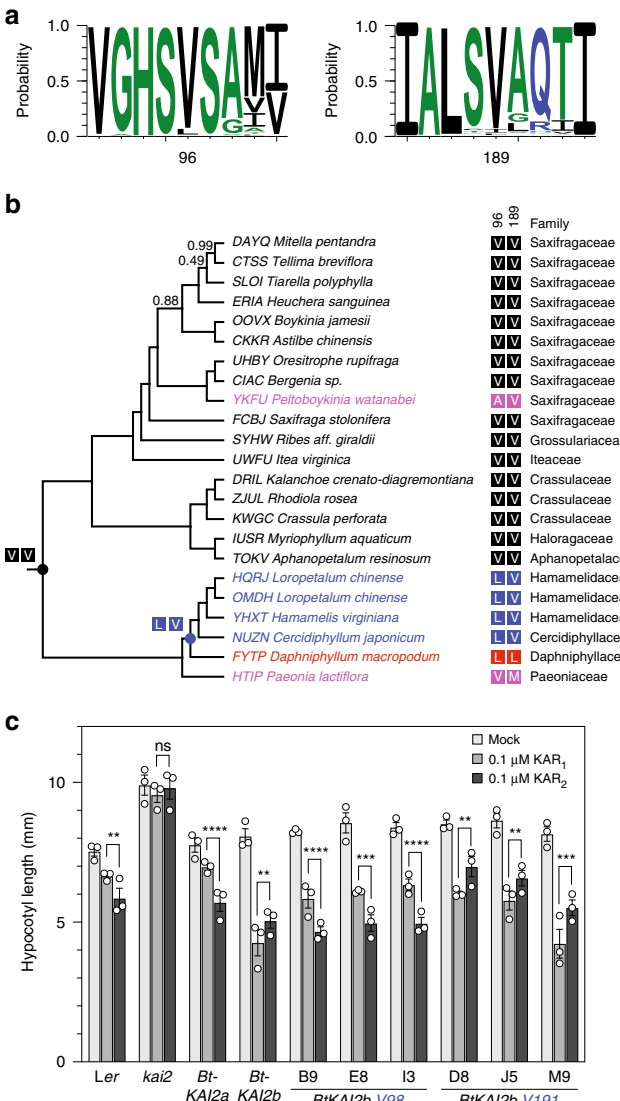

**Fig. 5 Diversity at positions 96 and 189 in KAI2 homologues across angiosperms. a** Sequence logos generated from 476 aligned KAI2 homologues sampled from eudicots, monocots and magnoliids and centred on position 96 (left) and 189 (right). **b** Species tree of the Saxifragales based upon 410 single-copy nuclear genes estimated using the ASTRAL gene tree method[50]. Branch support values are shown only where <1. The four-letter taxon prefix corresponds to the 1000 Plant Transcriptomes project Sample ID. Taxa are highlighted by colour according to identity of amino acids of KAI2 orthologues at positions 96 and 189, respectively: black, VV; blue, LV; red, LL; magenta, other. Postulated ancestral identities at specific nodes are indicated with filled circles. **c** Hypocotyl elongation responses to karrikins in three independent transgenic lines homozygous for GFP-BtKAI2a[L98;V191] and three lines for GFP-BtKAI2b[V98;L191]. Data shown are a summary of three experimental replicates performed on separate occasions, each comprising ~20 seedlings per genotype/treatment combination. Data from each replicate are presented in Supplementary Fig. 13. Error bars are SE, $n = 3$ experimental replicates; each dot corresponds to the mean value derived from each replicate. Asterisks denote significant differences: $*P < 0.05$, $**P < 0.01$, $***P < 0.001$, $****P < 0.0001$ (linear mixed model with experimental replicate as a random effect; specific pairwise comparisons using Tukey's HSD correction). Source data are provided as a Source Data file.

homologues stems from crystallography and in vitro binding assays. Two crystal structures of KAR-responsive KAI2 proteins from *Arabidopsis* and the parasitic plant *Striga hermonthica* reveal largely similar overall protein structure, but surprisingly are non-congruent with respect to $KAR_1$-binding position and orientation[52,53]. The affinity of AtKAI2 for $KAR_1$ is imprecisely defined, with estimates of dissociation coefficients (Kd) ranging from 4.6 μM[54] to 148 μM[55] using isothermal calorimetry, and 9 μM using fluorescence micro-dialysis[52]. Although variability in affinity estimates can be explained in part by different experimental conditions and techniques, it should also be considered that, depending on the homologue under examination, $KAR_1$ may not be the optimal ligand. Furthermore, binding of a hydrophobic compound to a hydrophobic pocket does not necessarily equate to ligand recognition or receptor activation. Our data, which are derived from clear and distinct biological responses, provide strong evidence that KAI2 is sufficient to determine ligand specificity, which, in turn, strengthens the case that KAI2 is the receptor by which karrikins are perceived.

The predominance of valine and leucine at position 96 (98 in BtKAI2 proteins), observed in 99.2% of our sample of 476 KAI2 sequences, suggests that this site is critical to protein function, which makes sense given its immediate proximity to the catalytic serine. This position not only contributes to ligand preference, but also to protein stability, given the negative effects of R98 in BtKAI2c. Compared with valine and leucine, arginine is relatively bulky with a charged side chain; indeed, in the few KAI2 sequences in which position 96 is neither leucine nor valine, the residue is nevertheless small and hydrophobic (either methionine or alanine; Supplementary Table 3). Interestingly, substitutions in AtKAI2 at or near the catalytic site, such as S95F, S119F and G245D, are not tolerated and render the protein unstable in plants[43]. The likely signalling mechanism for KAI2—by analogy of that for D14—involves ligand binding and/or hydrolysis, which triggers conformational changes in the lid domain and formation of a protein complex with MAX2 and SMAX1/SMXL2 protein partners[30,31,56]. As a result of signalling, KAI2 itself is degraded[40,43]. It is possible that KAI2 is an inher-ently metastable protein that is prone to conformational change and subsequent degradation by necessity; perhaps, this explains why it is so susceptible to instability through mutation.

Ligand specificity is a key contributor to the functional dis-tinction between karrikin and strigolactone receptors, and eluci-dating the molecular mechanisms behind ligand specificity is a significant research challenge. In certain root-parasitic weeds in the Orobanchaceae, substitutions of bulky residues found in AtKAI2 with smaller hydrophobic amino acids that increase the ligand-binding pocket size have likely improved the affinity for host-derived strigolactone ligands, as opposed to the smaller-sized karrikin-type ligands[57–59]. Moreover, lid-loop residues that affect the rigidity and size of the ligand entry tunnel determine the ligand selectivity between $KAR_1$ and *ent*-5-deoxystrigol (a strigolactone analogue with non-natural stereochemistry) among eleven KAI2 homologues in *Physcomitrella patens*[60]. Karrikin and strigolactone compounds also show chemical diversity within themselves, yet ligand discrimination among KAI2 and D14 receptors is not well characterised. Our data demonstrate that, in the case of BtKAI2 proteins, the two KAI2 homologues respond differently to subtly different KAR analogues, and that subtle changes in pocket residues likely account for preferences between these ligands. Position 96 appears to play a primary role in determining ligand specificity, whereas position 189 may improve ligand sensitivity, protein stability, or protein conformational dynamics following a prior mutation at position 96. Although the identified residues L98 and L191 are not predicted to change

dramatically the pocket size of BtKAI2b in comparison with BtKAI2a, these residues would make the BtKAI2b pocket more hydrophobic, which is consistent with $KAR_1$ being more hydrophobic than $KAR_2$[4]. Therefore, ligand specificity between highly similar chemical analogues can be achieved through fine-tuning of pocket hydrophobicity. Similarly, subtle changes have been reported for the rice gibberellin receptor GID1: changing Ile133 to leucine or valine increases the affinity for $GA_{34}$ relative to the less-polar $GA_4$, which lacks just one hydroxyl group[61].

It is possible that what we learn about ligand specificity in KAI2 proteins may also be informative about strigolactone perception by D14. Different D14 proteins may show ligand specificity towards diverse natural strigolactones, resulting in part from multiplicity of biosynthetic enzymes, such as the cytochrome P450 enzymes in the MAX1 family[45,62,63] and supplementary enzymes such as LBO[64]. Although the functional reasons underlying such strigolactone diversity are still unclear, it is possible that variation among D14 homologues yields varying affinities for different strigolactone analogues[65]. As an increasingly refined picture emerges of the features that determine ligand specificity for KAI2 and D14 receptors, we envision the rational design of synthetic receptor proteins with desirable ligand specificity in the future.

Gene duplication is a common feature in plant evolutionary histories as an initial step toward neofunctionalisation[66]. In obligate parasitic weeds, duplication and diversification of KAI2 genes (also referred to as HTL genes) have shifted ligand specificity towards strigolactones[57,58]. Karrikins, as abiotic molecules with limited natural occurrence, are unlikely to be the natural ligand for most KAI2 proteins, which are found throughout land plants. Instead, the evolutionary maintenance of a highly conserved receptor probably reflects the core KAI2 function of perceiving an endogenous ligand (KL) that regulates plant development[34–36]. This core function is presumably retained after gene duplication events through original, non-divergent copies that are under purifying selection[57,58]. KAI2 diversity may also reflect differing affinity for KL variants in different species and at different life stages. Our results are consistent with a scenario in which both BtKAI2a and BtKAI2b have retained the function of perceiving KL, because both copies complement the Arabidopsis kai2 phenotype, especially at the seedling and later stages. However, BtKAI2b has also acquired mutations that alter its ligand specificity, which in turn enhance sensitivity to the archetypal karrikin first discovered in smoke. The BtKAI2b gene is also more highly expressed than BtKAI2a in seeds and seedlings, consistent with an adaptation to post-fire seedling establishment. This diversity among KAI2 proteins may provide B. tournefortii with a selective advantage in fire-prone environments, contributing to its invasive nature.

We have shown evidence for the independent evolution of valine-to-leucine substitutions in KAI2 homologues throughout the dicots, albeit at low frequency, suggesting that these changes may have adaptive significance in various ecological contexts. It will be of interest to assess the function of such homologues in other species, and to validate and extend the specific conclusions we have made here. Recent findings indicate that KAI2 is an integrator that also modulates germination in response to other abiotic environmental signals, including temperature and salinity[67]. As germination is a critical life stage for seed plants, strategies for environmental conservation, restoration and weed control will benefit from specific knowledge of KAI2 sequence diversity and expression profiles.

## Methods

**Chemical synthesis.** Karrikins ($KAR_1$ and $KAR_2$), $^{13}[C]_5$-labelled karrikins and GR24 enantiomers ($GR24^{5DS}$ and $GR24^{ent-5DS}$) were prepared as previously described[68,69].

**Plant material and measurement of growth.** Arabidopsis kai2-2 (Ler) and Atd14-1 (Col-0) mutants were previously described[20]. Arabidopsis plants were grown on a 6:1:1 mixture of peat-based compost (Seedling Substrate Plus; Bord Na Mona, Newbridge, Ireland), vermiculite and perlite, respectively. Light was provided by wide-spectrum white LED panels emitting 120–150 µmol photons $m^{-2} s^{-1}$ with a 16 h light/8 h dark photoperiod, a 22 °C light/16 °C dark temperature cycle, and constant 60% relative humidity. The seeds of B. tournefortii used in this work were collected in November and December 2009 from two sites in Western Australia (Kings Park in Perth, and Merridin)[70]. B. tournefortii seeds were dried to 15% relative humidity for 1 month prior to storage in air-tight bags at −20 °C.

Arabidopsis rosettes were photographed at ~3–4 weeks after germination, prior to visible flowering, as specified in the figure legends. Plant height and shoot branching were determined once growth of the primary inflorescence had ceased. Height was measured using a ruler from the base of the primary inflorescence to the apex. Primary shoot branches were counted by inspecting the axils of each leaf; a bud was counted as a primary branch if it had elongated more than 5 mm.

**Seed germination assays.** Arabidopsis seeds were harvested from three pools of four plants per genotype, grown simultaneously under identical conditions (~120–150 µmol photons $m^{-2} s^{-2}$ white light, 16 h light/8 h dark, 22 °C light/16 °C dark). After harvesting, seeds were dried under silica gel for four days, and stored at −80 °C. Seeds were sown on 60 mm petri dishes containing 1% Phytagel solidified with 1.5 mM $MgCl_2$ and supplemented with 0.1% v/v acetone (mock) or 1 µM $KAR_1/KAR_2$ (1:1000 dilution). Petri dishes were incubated at 25 °C under constant light for up to 1 week, and germination was scored using a dissecting microscope every 24 h. B. tournefortii seeds were sown in triplicates (35–70 seeds each) on glass microfibre filter paper (Grade 393; Filtech, NSW Australia) held in 9-cm petri dishes and supplemented with mock or karrikin treatments (3 mL of aqueous treatment solution per petri dish). Treatments were prepared by diluting acetone (mock) or karrikin stocks (dissolved in acetone) 1:1000 with ultrapure water. Because germination of B. tournefortii is inhibited by light[9], the seeds were imbibed in the dark at 22 °C. Numbers of germinated seeds were counted each day until the germination percentages remained unchanged.

**Hypocotyl elongation assays.** Approximately 50 Arabidopsis seeds were sown on 60 mm petri dishes containing half-strength MS salts (pH 5.9) and 0.7% w/v agar. The plates were stratified in the dark at 4 °C for 72 h, transferred to a growth incubator and exposed to white light at 22 °C for 3 h, and then to darkness for 21 h at 22 °C. The plates were then exposed to continuous red light supplied by light-emitting diodes (LEDs) (Philips GreenPower LED Research module; $λ_{max}$ 660 nm, 5 µmol photons $m^{-2} s^{-1}$) for 4 days at 22 °C. Seedlings were laid out horizontally on 0.7% w/v agar for photography, and hypocotyl lengths were measured using ImageJ software (https://imagej.nih.gov/ij/index.html). For B. tournefortii, assays were performed with the following modifications to the Arabidopsis protocol: B. tournefortii seeds were sown on glass microfibre filter paper (Filtech) held in petri dishes supplemented with mock or karrikin treatments. The seeds were imbibed in the dark for 22 h at 24 °C before exposing to continuous red light (5 µmol photons $m^{-2} s^{-1}$) for 4 days, at which point the hypocotyls were measured as above.

**Transcript analysis.** B. tournefortii seeds were imbibed on glass fibre filters in the dark at 22 °C and treated with karrikins using the same procedure as described under seed germination assays, above. For transcript quantification in B. tournefortii seeds, samples were taken at the indicated time points, blot-dried on paper towel, and frozen in liquid nitrogen. For B. tournefortii seedlings, two methods were used. For data shown in Fig. 1e, seeds were sowed and seedlings grown in triplicate under identical conditions to those described for hypocotyl elongation assays. For data shown in Fig. 2b, seeds were first imbibed for 24 h in the dark, and then transferred to the growth room (~120–150 µmol photons $m^{-2} s^{-2}$ white light, 16 h light/8 h dark, 22 °C light/16 °C dark) for 3 days. A sample of ~20 seedlings was then transferred to a 250 mL conical flask containing 50 mL sterile ultrapure water supplemented with 1 µM $KAR_1$, or an equivalent volume of acetone (0.1% v/v), with three replicate samples per treatment. The flasks were then shaken for 24 h before seedlings were harvested.

Arabidopsis seedlings were grown on $0.5 × MS$ agar under growth room conditions (wide-spectrum white LED panels emitting 120–150 µmol photons $m^{-2} s^{-1}$ with a 16 h light/8 h dark photoperiod and a 22 °C light/16 °C dark temperature cycle) for 7 days. On the 7th day, seedlings were transferred to 3 mL liquid $0.5 × MS$ medium in 12-well culture plates (CORNING Costar 3513) and shaken at 70 rpm for a further 22 h under the same growth room conditions, after which the medium was removed by pipette and replaced with a fresh medium containing relevant compounds or an equivalent volume (0.1% v/v) of acetone. After further incubation with shaking (8 h), the seedlings were harvested, blotted dry, and frozen in liquid nitrogen. Dry Arabidopsis seeds (20 mg) were transferred to a 2-mL tube and imbibed in 1 mL of water, supplemented with karrikins or an equivalent volume of acetone, and incubated with end-over-end rotation for 72 h in the dark at 4 °C. Seeds were transferred to the light (16 h light/8 h dark, 120–150 µmol photons $m^{-2} s^{-1}$) for at 22 °C for a further 24 h with rotation. Seeds were harvested by brief centrifugation, removal of excess water, and freezing in liquid nitrogen and were ground to a fine powder in a pestle and mortar.

**Transcript quantification**. RNA extraction was performed using the Spectrum Plant Total RNA kit (Sigma-Aldrich) with an on-column DNase digestion step. cDNA was generated from 500 ng total RNA using the iScript cDNA Synthesis kit (Bio-Rad), and reactions were subsequently diluted fourfold in water. Quantitative PCR was conducted in 5-microlitre reactions in a 384-well format using a Roche LightCycler 480. Reactions contained 0.4 µM oligonucleotides, 0.5 µl diluted cDNA and 1× Luna Universal qPCR Master Mix (New England Biolabs). Two technical replicates of each reaction were performed for each biological replicate, and the mean Cp value was used to normalise expression to *CACS* reference transcripts using the formula $(1 + E_{gene})^{-\text{Cp\_gene}}/(1 + E_{CACS})^{-\text{Cp\_CACS}}$, where $E$ is the primer efficiency. Primer efficiencies (0.9 or greater) were determined in separate runs using serial dilutions of pooled cDNA. All oligonucleotides are listed in the Supplementary Table 3.

**Cloning and mutagenesis of *KAI2* and *D14* homologues**. Full-length *BtKAI2*-coding sequences (and unique 3′-UTR sequences) were amplified from *B. tournefortii* cDNA with Gateway-compatible *attB* sites using the universal forward primer BtKAI2_universal_F and homologue-specific reverse primers RACE_R (BtKAI2a), Contig1_R (BtKAI2b) and Contig5_R (BtKAI2c), before cloning into pDONR207 (Life Technologies). The pDONR207 clones were confirmed by Sanger sequencing and recombined with pKAI2pro-GFP-GW[41] to generate the binary plant expression plasmids pKAI2pro-GFP-BtKAI2a, pKAI2pro-GFP-BtKAI2b and pKAI2pro-GFP-BtKAI2c. For transient over-expression in tobacco (Supplementary Fig. 5d), BtKAI2-coding sequences were transferred via Gateway-mediated recombination into pSKI106, which drives very high expression of coding sequences placed immediately downstream of a full-length cDNA clone of the tobacco mosaic virus U1 strain, all under control of the CaMV 35 S promoter in a standard T-DNA binary vector[42]. pSKI106 also encodes an N-terminal 3 × c-myc tag. For transient expression in tobacco using standard binary vectors (Supplementary Fig. 6b), the *BtKAI2* coding sequences were transferred into pMDC43, which is identical to pKAI2pro-GFP-GW except that the *AtKAI2* promoter & 5′-UTR sequences are replaced with a double CaMV 35 S promoter[71]. The *BtD14a*-coding sequence was amplified from cDNA using oligonucleotides BtD14_F and BtD14_R, cloned into pDONR207 as above, and transferred into pD14pro-GW[41].

The full-length *BtKAI2*-coding sequences (excluding 3′-UTRs) were amplified from the pDONR207 clones and reconstituted with the pE-SUMO vector by Gibson Assembly to generate the heterologous expression plasmids pE-SUMO-BtKAI2a, -BtKAI2b and -BtKAI2c. Site-directed mutagenesis generated pE-SUMO-BtKAI2a$^{L98;L191}$, pE-SUMO-BtKAI2b$^{V98;V191}$ and the BtKAI2c variants. For expression of mutated versions of BtKAI2a, BtKAI2b, BtKAI2c, and AtKAI2 in tobacco and *Arabidopsis*, site-directed mutagenesis was performed on pDONR207 clones prior to recombination with pKAI2pro-GFP-GW or pMDC43. In the case of the double substitutions in BtKAI2a and BtKAI2b, both targeted residues were mutated simultaneously in one PCR product, whereas the remainder of the plasmid was amplified in a second PCR product. These double mutated plasmids were reconstituted by Gibson assembly. For the single mutants, the entire plasmid was amplified using a single pair of mutagenesis primers, and the PCR product was self-ligated. The BtKAI2c$^{QUINT}$ variant was generated by two successive rounds of mutagenesis. Coding regions were confirmed by Sanger sequencing.

**Plant transformation**. Homozygous *kai2-2* and *Atd14-1* plants were transformed by floral dip. Primary transgenic seedlings were selected on sterile 0.5 × MS medium supplemented with 20 µg/mL hygromycin B. $T_2$ lines exhibiting a 3:1 ratio of hygromycin resistant-to-sensitive seedlings were propagated further to identify homozygous lines in the $T_3$ generation. Experiments were performed from the $T_3$ generation onwards.

For transient expression in tobacco, *Agrobacterium* (GV3101) carrying pSKI106 or pMDC43 variants was grown in LB medium (25 mL) supplemented with antibiotics and 20 µM acetosyringone until $OD_{600}$ reached 1.0. The bacteria were then harvested by centrifugation (15 min, 5000 × g) and resuspended in 10 mM MgCl₂, 10 mM MES (pH 5.6) and 100 µM acetosyringone. The optical density was adjusted to 0.4, and the suspension was left standing at 22 °C overnight (~14 h). Leaves of 3-week-old *Nicotiana benthamiana* were then infiltrated with a 5-mL syringe, through the abaxial leaf surface. After 4 days, the leaves were collected and frozen in liquid nitrogen.

**Immunoblotting and antibodies**. Seedlings were frozen in liquid nitrogen and ground to a fine powder using a bead mill. Soluble proteins were extracted with 150 µl of lysis buffer (50 mM TRIS-HCl pH 7.5, 150 mM NaCl, 10% glycerol, 0.1% Tween-20, 1 mM phenylmethylsulfonyl fluoride, 1 mM dithiothreitol (DTT) and 1 × complete protease inhibitors (Roche)) per 100 mg of tissue. Lysates were clarified by two successive centrifugations at 20,000 × g for 10 minutes each. Approximately 40 µg protein was separated on a 12% SDS-PAGE gel and blotted onto PVDF membrane. Blots were blocked in 1 × TBS-T (pH 7.5) and 2% (w/v) bovine serum albumin (BSA) for 1 h at 22 °C. Primary antibodies were diluted in 1 × TBS-T + 0.2% (w/v) BSA as follows: 1:2500 (anti-KAI2 raised in rabbit[40]); 1:1000 (anti-GFP raised in rabbit, ThermoFisher Scientific A11122), 1:2500 (anti-actin raised in mouse, Sigma A0480) and 1:1000 (anti-c-myc raised in mouse, Genscript A00704). Primary antibodies were incubated with the blot either overnight at 4 °C, or for 1 h at 22 °C, with gentle rocking. Between antibody incubations, blots were washed four times (5 min per wash) with 1 × TBS-T. Secondary antibody (goat anti-rabbit IgG-HRP conjugate, ThermoFisher 32460; or goat anti-mouse IgG-AP conjugate, ThermoFisher 31321) was diluted 1:1000 (HRP) or 1:5000 (AP) in 1 × TBS-T and 0.2% (w/v) BSA, and incubated with the blot for 1 h at 22 °C. HRP activity was detected using Clarity Western ECL substrate (Bio-Rad), and AP activity was detected with ImmunStar AP substrate (Bio-Rad). Chemiluminescence images (16-bit) were collected using the ImageQuantRT ECL system (GE Healthcare).

**Karrikin uptake measurements**. Fifteen samples of seeds were sowed for each karrikin treatment (five time points, each in triplicate). In each sample, ~40 mg of *B. tournefortii* seeds were imbibed in 3 mL ultrapure water for 24 h in 5-mL tubes. After centrifugation (2 min at 3220 × g), excess water was removed by pipette and the volume of residual water (mostly absorbed into the seeds) was calculated by weighing the seeds before and after imbibition. Fresh ultrapure water was added to the fully imbibed seeds to reach a total volume of 980 µL. Then 20 µL of 100 µM KAR₁ or KAR₂ was added to a final concentration of 2 µM. The seeds were imbibed at 22 °C in darkness. At the indicated time point (0, 2, 4, 8 or 24 h post treatment), 500 µL of the imbibition solution was removed and combined with 100 ng of either$^{13}[C]_5$-KAR₁ or $^{13}[C]_5$-KAR₂ (100 µL at 1 µg/mL) as an internal standard for quantification purposes. The sample was then extracted once with ethyl acetate (500 µL), and 1 µL of this organic layer was analysed using GC-MS in selective ion monitoring (SIM) mode as previously described[72].

The amount of KAR₁ in each sample was calculated by the formula $\frac{A(Ion150)}{A(Ion155)} \times 100\,ng$ and converted to moles, where $A(Ion150)$ indicates the peak area of the ion 150 (KAR₁ to be measured), $A(Ion151)$ indicates the peak area of the ion 151 ($^{13}[C]_5$-KAR₁), and 100 ng is the amount of $^{13}[C]_5$-KAR₁ spiked in before the ethyl acetate extraction. Similarly, the amount of KAR₂ in each sample was calculated by the formula $\frac{A(Ion136)}{A(Ion141)} \times 100\,ng$ and converted to moles. The uptake percentage adjusted to 40 mg of *B. tournefortii* seeds was calculated by the formula: $\left(1 - \frac{N(x)}{N(0)}\right) \times \frac{40mg}{m(seed)}$, where $N(0)$ indicates moles of karrikins at time 0, $N(x)$ indicates moles of karrikins at time point $x$, and $m(seed)$ indicates the dry weight (mg) of seeds tested in each replicate. For *Arabidopsis* seeds, the procedure was scaled down for a smaller mass of seeds (20 mg).

**Transcriptome assembly and analysis**. Twenty milligrams of dry *B. tournefortii* seeds were imbibed in water for 24 h and incubated at 22 °C in the dark. The seeds were collected by centrifugation, blotted dry and frozen in liquid nitrogen. A separate sample of seeds was sown on glass filter paper, imbibed for 24 h as above, and then incubated for 96 h under continuous red light (20 µmol m⁻² s⁻¹) with a 22 °C (16 h)/ 16 °C (8 h) temperature cycle. A single sample of seedlings (50 mg fresh weight) was harvested and frozen in liquid nitrogen. Total RNA was extracted from both seed and seedling samples using the Spectrum Plant RNA kit (Sigma-Aldrich), including an on-column DNase step. PolyA⁺ mRNA was purified using oligo(dT) magnetic beads (Illumina), and cDNA libraries for sequencing were generated as described[73]. Sequencing was performed on the Illumina HiSeq 2000 platform at the Beijing Genomic Institute, Shenzhen, China. Raw reads were filtered to remove adapters and low-quality reads (those with >5% unknown nucleotides, or those in which >20% of base calls had quality scores ≤10). After filtering, both libraries generated reads with >99.9% of nucleotides attaining Q20 quality score. Transcriptome de novo assembly was performed with Trinity[74]. For each library, contigs were assembled into Unigenes; Unigenes from both libraries were then combined, yielding a total of 45,553 predicted coding region sequences with a mean length of 1011 nt. The combined Unigenes were then interrogated for homology to *AtKAI2*, *AtD14*, *AtDLK2*, *AtSTH7* and *AtCACS* using BLASTn searches.

**Phylogenetic analysis and protein sequence analysis**. *KAI2* and *D14* homologues in *Brassica* species were identified from BLAST searches using *Arabidopsis* coding sequences as a query. Additional sequences were sampled from an existing phylogenetic analysis[33]. Multiple sequence alignments were performed using MAFFT plugin implemented in Geneious R10 (Biomatters). The alignment was trimmed slightly at the 5′-end to remove non-aligned regions of monocot D14 sequences. Maximum likelihood phylogenies were generated using PHYML (GTR + G + I substitution model, combination of NNI and SPR search, and 100 bootstraps). The choice of substitution model was guided by Smart Model Selection in PhyML[75] (http://www.atgc-montpellier.fr/sms). A list of all sequences, and their sources, is provided in Supplementary Table 1.

For analysis of BtKAI2c using PROVEAN, all 10 positions that were unique to BtKAI2c when compared with AtKAI2 and other *Brassica* sequences (Supplementary Fig. 3) were manually edited to the consensus identity at each position. This was the 'reverted' sequence against which each of the native BtKAI2c amino-acid substitutions were tested using the PROVEAN server (http://provean.jcvi.org) using the default cutoff score of −2.5. The PROVEAN output is presented in Supplementary Table 2.

*KAI2* homologues from The 1000 Plant Transcriptomes Project[50] were identified by TBLASTN searches using *A. thaliana* KAI2 protein sequence as a query against 'onekp database v5', which comprises 1328 individual samples (https://db.cngbdb.org). The search was configured to return a maximum of 500

target sequences with expect values <0.01, using a word size of six and the BLOSUM62 scoring matrix. Nucleotide sequences were collated, and open reading frames (ORFs) >700 nucleotides were predicted in Geneious software. A single ORF was selected for each sequence; those with multiple ORFs were screened manually for the one with homology to AtKAI2. The ORFs were translated and aligned using the MAFFT multiple alignment tool. A few misaligned or mis-translated sequences were removed from the alignment manually, resulting in a data set of 476 unique sequences. Extracted parts of this alignment centred on positions 96 and 189 were used to generate WebLogos (http://weblogo.threeplusone.com[76]). A list of identified KAI2 homologues and their positional analysis are presented in Supplementary Data 1.

**Protein homology modelling**. KAI2 structures were modelled using the SWISS-MODEL server (https://swissmodel.expasy.org) using the alignment mode[77] and the *A. thaliana* KAI2 structure 3w06 as a template[54]. Figures of protein structure and homology models were generated using PyMOL v1.3 (Schrödinger LLC). Cavity surfaces were visualised using the 'cavities & pockets (culled)' setting in PyMOL and a cavity detection cutoff value of four solvent radii. Cavity volumes were calculated using the CASTp server v3.0[78] (http://sts.bioe.uic.edu/castp) with a probe radius of 1.4 Å. Values indicate Connolly's solvent-excluded volumes. Cavities were inspected visually using Chimera v1.12 (https://www.cgl.ucsf.edu/chimera/). For both BtKAI2a and BtKAI2b models, CASTp erroneously included surface residues near the primary pocket entrance in the calculation of the pocket volumes. This issue was resolved by the artificial placement of a free alanine residue adjacent to the cavity entrance, as described previously[79].

**Protein expression and purification**. AtKAI2 and BtKAI2-coding sequences were cloned into pE-SUMO-Amp (LifeSensors) to generate N-terminal 6 × HIS-SUMO fusion proteins. All proteins were expressed in BL21 Rosetta DE3 pLysS cells (Merck-Millipore) and purified using IMAC as described in detail previously[41]. In brief, cultures were grown in Luria-Bertani medium at 30 °C until the optical density reached 0.8–1, at which point the cultures were chilled to 16 °C and induced with 0.1 mM isopropyl β-D-1 thiogalactopyranoside and allowed to grow for a further 14–16 h. Pellets were lysed in 20 mM HEPES pH 7.5, 150 mM NaCl, 10% glycerol, 10 mM imidazole and 1 × BugBuster reagent (Merck-Millipore). Proteins were purified from lysates in batch mode using gravity columns containing TALON cobalt affinity resin (Takara). Columns were washed with 10 mM imidazole and eluted with 200 mM imidazole in 1-mL fractions. Proteins were concentrated and imidazole removed by ultrafiltration and buffer exchange into 20 mM HEPES pH 7.5, 150 mM NaCl, 10% glycerol.

**Differential scanning fluorimetry**. DSF was performed in 384-well format using a LightCycler 480 instrument (Roche). Reactions (10 µL) contained 20 µM protein, 20 mM HEPES pH 7.5, 150 mM NaCl, 1.25% (v/v) glycerol, 5 × SYPRO Tangerine dye (ThermoFisher Scientific) and varying concentrations of ligand that resulted in a final concentration of 5% (v/v) acetone. The programme parameters were: excitation at 483 nm, emission at 640 nm, ramp rate of 0.2 °C/s from 20 to 80 °C. The first derivative of fluorescence over temperature was calculated using the 'Tm calling' function of the LightCycler software. Raw data from technical replicates were averaged and plotted using GraphPad Prism 8.1 (GraphPad Software, graphpad.com).

**Statistical analysis**. Data were analysed using one- or two-way analysis of variance (ANOVA) ($\alpha = 0.05$, with Tukey's multiple comparisons test). For Figs. 4c and 5c, in which data from three experimental replicates were combined, data were analysed using a mixed effects model with experimental replicate as a random effect, and genotype and treatment as fixed effects. Prior to ANOVA, germination data were arcsine-transformed, and gene expression data were log-transformed. Tests were implemented in GraphPad Prism version 8.1.

**Reporting summary**. Further information on research design is available in the Nature Research Reporting Summary linked to this article.

## Data availability

All relevant data and materials are available from the authors upon reasonable request. Sequence data are available at NCBI Genbank under the following accessions: *BtKAI2a*, MG783328; *BtKAI2b*, MG783329; *BtKAI2c*, MG783330; *BtD14a*, MG783331; *BtD14b*, MG783332; *BtDLK2*, MG783333; *BtSTH7*, MK756121; *BtCACS*, MK756122. Raw RNA sequence data from *B. tournefortii* seed and seedlings are available in the NCBI SRA database under accession SRP128835. The original data underlying the following figures are provided as a Source Data file: Fig. 1a, b, d, e; 2b, e; 3a–d; 4a–d; 5a–c; Supplementary Figs. 1b–f; 2; 3; 4c; 5b–d; 6b, c, e; 7; 8a–e; 9a–c; 10a; 11; 12; 13.

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

## Acknowledgements

This work was supported by funding from the Australian Research Council (DP130103646 to S.M.S. and G.R.F.; DP140104567 to S.M.S.; FT150100162 to M.T.W., DP160102888 to G.R.F. and M.T.W.). Y.K.S was recipient of a Research Training Program Scholarship from the Australian Government. We thank Dr Rowena Long for the original provision of *B. tournefortii* seeds, Dr Rohan Bythell-Douglas for advice on homology modelling and Mr Yongjie Meng with assistance with RNA extractions. We are grateful to Dr Kevin Rozwadowski (Agriculture and Agri-Food Canada) for the provision of pSKI106.

## Author contributions

Y.K.S., S.M.S., C.S.B., G.R.F. and M.T.W. conceived and designed the research. Y.K.S. and M.T.W. performed the majority of experiments with assistance from J.Y. (hypocotyl elongation assays and screening of transgenic lines), A.S. (chemical synthesis and seed germination assays), K.T.M. and S.F.D. (protein expression and purification). Y.K.S. and M.T.W. analysed the data. Y.K.S., S.M.S. and M.T.W. wrote the manuscript.

## Competing interests

The authors declare no competing interests.
