## [Peer Review File · Nature Communications]

Reviewers' comments:

Reviewer #1 (Remarks to the Author):

Remarks to the Author

In the last few years, there has been substantial progress in understanding the karrikin signaling pathway particularly about how karrikins are selected and perceived by their receptors. The manuscript by Sun et al. characterize KAI2 homologs in *Brassica tournefortii*. Remarkably, when individually expressed in *Arabidopsis*, the receptors show distinct responses to karrikins KAR1 or KAR2. The authors use theoretical and experimental analyses to identify and describe two distinct amino acids that confer specificity to each of the ligands. Overall, the identification of these two residues is novel in the field of karrikin signaling. Yet the study can be more impactful to the wider research community if the authors can provide data beyond Brassicaceae to better ascertain the evolutionary significance of their findings.

Major points:

1. Since most of the findings underline the effect of karrikins on seedlings, it would be helpful if the Introduction included more background on the importance of karrikin signaling during seedling development.
2. As a general comment, I found some of the concluding statements over speculative and/or not fully supported by results. Specific examples:
 - a. Lines 74-77: the data presented in the manuscript is limited to Brassicaceae, and the claim that the findings can be extrapolated to "broader family of KAI2-D14 receptor-enzyme" has not been demonstrated.
 - b. Lines 100-102: the conclusion made by the author about the ancestral state of KAI2 gene copies is inaccurate given Figure 2a and S2 (see point 4)
 - c. Lines 126-128, 191-193: it is questionable whether failure to detect KAI2c directly correlates to its functionality. BtKAI2c is expressed in seeds higher than BtKAI2a, which raises a question of its sensing function and transcriptional regulation in seeds. Moreover, driving expression with AtKAI2 promoter may not fully recapitulate native functional patterns, and poor levels of BtKAI2c proteins via transient expression in tobacco leaves (Figure S5c) does not necessarily reflect native protein levels (see point 3). Lastly, BtKAI2c functionality was not tested by DSF, and its solubility was not assessed therefore the conclusion is weakly supported.
3. The early conclusion regarding BtKAI2c led to insufficient analyses and perhaps a missed opportunity to parse apart the characterization of the receptor. Addressing the receptor potential with additional regulatory elements such as 35S:BtKAI2 and/or pUBQ:BtKAI2, might be more revealing. As mentioned above, the conclusion that BtKAI2c is not functional requires additional data. Please refer to Burger et al 2019, where transgenic *Arabidopsis* expressing PpKAI2 homologs show no complementation but were able to bind karrikin KAR1 in vitro.
4. Figure 2a and S2a: generation of tree from highly similar sequences introduces uncertainty into the validity of the conclusion made about BtKAI2 ancestry. Also, underrepresentation of plant diversity further weakens the author's conclusion (in line 100-102). If any ancestry can be inferred, the length of branches indicates KAI2b is more similar to the outgroup, thus the conclusion that KAI2a is more ancestral is unsupported. The phylogenetic tree is in agreement with BtKAI2a's similarity to AtKAI2 and its splitting from BtKAI2b and BtKAI2c. Broadening the survey of sequences could provide better support for claims about ancestry.
5. While the complementation experiments in plants drive the main conclusion of this study, Figure 4 doesn't provide compelling evidence. The use of GR24ent-5DS, a non-natural and unrelated ligand for KAI2, may belie the importance of the residues in conferring karrikin selectivity to BtKAI2s. Given that KAI2s respond poorly to GR24 (or slightly respond to very high concentrations of GR24, as in Figure S9b), the overall effect and statistics (Figure 4a, f) are very mild and perhaps even insignificant if all tests would be scaled to one another. I am wondering if the authors will be able to test the three homologs and the swapped mutants in alternative assays

such as ITC or Microdialysis (recently shown with karrikin in Xu et al 2016 and Burger et al 2019). Moreover, the structural modelling (Figure 4b-e) provides little information beyond speculation. The predicted changes in the pocket volume might indeed be attributed by the mild changes in amino acids, yet without a crystal structure (at least for BtKAI2b) or better binding assay, these predictions are very much suggestive. I do think it is important to help readers understand the rationale and the exact positions of the residues by 3D models, perhaps as supplementary data and not necessarily as main Figure. In that regard, Lopez-Obando et al 2016, provided similar structural prediction approach for PpKAI2's pockets that proven to be only partially true (Burger et al 2019).

6. It would be helpful to provide a broader in silico analysis in multiple species. This can strengthen the correlation of the two amino acid residues identified by the authors to KAR1 or KAR2 ligand preference.

Minor points:

1. Line 84: missing reference to justify the choice of target genes.
2. Detailed quantification of rescue experiments (whole seedling and leaf phenotype) is missing for Figure 2c-d. Full complementation is not supported by the data presented (Line 250), particularly those from germination assays. It would also be helpful if the authors reconciled the differences observed in degree of rescue presented in Figures 3a-b and also provided further statistical tests to ascertain the extent of rescue compared to Ler or *kai2-2*.
3. In order to better visualize and assess protein molecular weight and levels of expression, authors should provide uncropped gels as supplementary figures. In Figure 2e, the appearance of anti-actin blot and the wide size separation (between 30kD to 60kD on 12% PAGE) should be clarified.
4. I am wondering if it would be more comprehensive to include BtKAI2c in the differential expression experiment (Figure 3c) as was performed for Figure 3a-b.
5. DSF analyses were performed with His-SUMO tagged KAI2, this is probably related to the solubility state of un-tagged KAI2, but if so, why not testing SUMO-BtKAI2c. Some clarifications will help to better understand the usage of tagged protein in thermal shift assays.
6. In Figure 4b the label of V191 should be adjusted.
7. Line 323: please provide details on promoter used for transient expression besides the name of the vector.

Reviewer #2 (Remarks to the Author):

Brassica tournefortii is an invasive weed, easily colonizing the landscape after the wild fire due to its enhanced ability to respond to the major germination stimulant found in smoke, KAR1. In contrast, *Arabidopsis thaliana* is more responsive to less abundant KAR2. The authors proposed that *Brassica tournefortii* has three homologues of *Arabidopsis* karrikin receptor KAI2 and the most highly expressed one, BtKAI2b is sufficient to confer this enhanced response to KAR1. In addition, two amino acid residues were identified that are responsible for the ligand specificity of the receptor.

The manuscript is very interesting, presents a novel work and is clearly written. Understanding how ligand specificity of KAI2 can be determined is indeed of the high importance considering duplication of this receptor in many plant species.

Major remarks:

Line 122 and Supplementary figure 5c: *Arabidopsis* was transformed with GFP tagged BtKAI2s while the expression in tobacco was tested with myc-tagged protein. From the text it reads like the same constructs were tested in both systems. It would be more convincing to show the expression of GFP tagged proteins in tobacco. Other option is to demonstrate that there is no expression the

myc-tagged BtKAI2c in Arabidopsis and no complementation of kai2-2 mutant. There is quite some difference in the size of both tags so maybe this could influence the stability of BtKAI2c.

Line 129: BtKAI2a and BtKAI2b indeed show differential ligand specificity but only in seedlings. During seed germination BtKAI2a seems to be equally responsive to KAR1 and KAR2. The effect in seeds and seedlings should be clearly separated in the manuscript, because now it seems like BtKAI2a is always more responsive to KAR2, which is not the case. Also, the DLK2 expression was tested in seedlings, which is in agreement with the hypocotyl data, but does not help to understand the specificity, or its lack, of KAI2A during seed germination. I would suggest to check the DLK2 expression in the seeds of BtKAI2a and BtKAI2b lines in response to KAR1 and KAR2 treatment.

Line 189: It should be specified that this effect is observed only in seedlings. Additional germination assay would be strongly recommended to check if the KAR1 specificity of BtKAI2b can be reversed also in seeds of BtKAI2bV98;V191.

Supplementary figure 5: - The qPCR analysis should be done with three biological repeats not only one.

Minor remarks:

Line 66: The term "KL" should refer to KAI2 ligand not karrikin-like ligand, as there is no evidence, to my knowledge, that this unknown molecule is a karrikin-like.

Line 74: Using the term "KAI2-D14 family" in this sentence but also in throughout the text is confusing. First of all, the manuscript focuses on the KAI2 homologues and their ligand specificity not much is studied for D14. Secondly, DLK2 also belongs to this family but does not respond to strigolactone or karrikins.

Lines 83-86: There is no difference in the expression of both marker genes at 100 nM so stating that "ten-fold higher concentrations of KAR2 (1 μ M) compared with KAR1 (100 nM) was required to reach equivalent levels" is a bit unclear. I would rather describe the significant difference in marker gene expression that is clear when 1 μ M concentration is used.

Line 91: I suggest to rewrite the sentence as it sounds a bit strange that the expression was more responsive... The genes were more responsive or the expression was changed?

Line 141: This sentence is a bit overstatement. BtKAI2a is also responsive to KAR1 so BtKAI2b contributes to the enhanced KAR1-responsiveness in seeds rather than is fully responsible for it.

Line 181: It should be mentioned that in this case also KAI2 promoter was used.

Lines 293-295: It should be indicated how the treatment with karrikins was done in Brassica tournefortii seeds.

Figure 1b and Supplementary figure 1 and 7, line 375, 377, 436: the word "seed" should be in a plural form in several places. I suggest to check it carefully in the whole manuscript.

Figure 1c,d: Hypocotyl elongation responses of *B. tournefortii* seedlings treated with KAR1 and KAR2 and grown for four days under continuous red light.- I would rewrite the sentence to make clear that seedlings were grown for 4 days on the medium with KAR1 or KAR2.

Figure 2: Instead of "treated with or without 1 μ M KAR" I would rather write treated mock or with 1 μ M KAR.

Supplementary figure 2 and 3: The reference in figure legend should be changed to a number to fit

a reference style of the manuscript.

Supplementary figure 5: Consider rewriting the sentence: "RNA was isolated from approximately 50 seven-day-old seedlings per genotype, and are the same samples as those shown in the immunoblots in Figure 2."

Response to reviewers' comments

We thank the reviewers for constructive, helpful and respectful feedback. We believe that we have addressed most of their concerns and we are confident that the manuscript is much improved.

In summary, these are the main changes and additions made (in order of presentation in the text):

1. A thorough investigation into the reasons behind the lack of expression of BtKAI2c. In the previous version of the manuscript, we speculated that the R98 residue, which is very non-conservative, might be responsible for BtKAI2c instability. We tested this hypothesis directly by changing this residue back to a valine, and by doing so we restored expression in tobacco and stability in *E. coli*. We also generated the corresponding V96R mutation in AtKAI2, and this prevented expression in tobacco. These data are presented in Supplemental Fig 6 and described in an extensive section of the results.
2. Additional analysis of transcripts in BtKAI2a and BtKAI2b seeds (Fig 3b)
3. A closer analysis of the ligand specificity of BtKAI2s in seeds vs seedlings, including germination data from the swapped BtKAI2aL98;L191 and BtKAI2bV98;V191 transgenics (Supp Fig. 13)
4. An in silico analysis of sequence diversity at positions 96 and 189 using data from 1000 Plant Transcriptomes project (Fig 5)
5. Analysis of single amino acid substitutions in BtKAI2b, revealing that V98 is most probably responsible for the ligand preference (Fig 5 and Supp. Fig. 14)

Specific responses are as follows:

Reviewer # 1

Major points

1. Since most of the findings underline the effect of karrakins on seedlings, it would be helpful if the Introduction included more background on the importance of karrikin signaling during seedling development.

We agree, and have added a new paragraph in the introduction (second paragraph).

2. As a general comment, I found some of the concluding statements over speculative and/or not fully supported by results. Specific examples:

a. Lines 74-77: the data presented in the manuscript is limited to Brassicaceae, and the claim that the findings can be extrapolated to "broader family of KAI2-D14 receptor-enzyme" has not been demonstrated.

We have removed all references to the "KAI2-D14 family". We agree that our conclusions are limited to the proteins in question; we only intended to imply that our findings will have

potential relevance to strigolactone perception as well given that these proteins are all homologous in structure and function.

b. Lines 100-102: the conclusion made by the author about the ancestral state of KAI2 gene copies is inaccurate given Figure 2a and S2 (see point 4)

We have removed any claims of ancestry, which we agree is confusing, instead revising this section to state that BtKAI2a and AtKAI2 are more similar, and that BtKAI2b and BtKAI2c are paralogues that arose during genome triplication in Brassica. This is a parsimonious interpretation of the tree. It is not of great importance to our paper which copy of BtKAI2 is ancestral, just which one is more alike AtKAI2 and the KAI2s found as a single copy in other Brassicaceae. In any case, our new analysis in Fig 5 makes it quite likely that V96 V189 is the ancestral state (as is found in AtKAI2 and BtKAI2a).

c. Lines 126-128, 191-193: it is questionable whether failure to detect KAI2c directly correlates to its functionality. BtKAI2c is expressed in seeds higher than BtKAI2a, which raises a question of its sensing function and transcriptional regulation in seeds. Moreover, driving expression with AtKAI2 promoter may not fully recapitulate native functional patterns, and poor levels of BtKAI2c proteins via transient expression in tobacco leaves (Figure S5c) does not necessarily reflect native protein levels (see point 3). Lastly, BtKAI2c functionality was not tested by DSF, and its solubility was not assessed therefore the conclusion is weakly supported.

This concern has been fully addressed in a new section of the results, and in Supplementary Figure 6. We don't know why BtKAI2c transcripts are so high in Bt seeds but we presume that the encoded protein is probably non-functional in native plants.

3. The early conclusion regarding BtKAI2c led to insufficient analyses and perhaps a missed opportunity to parse apart the characterization of the receptor. Addressing the receptor potential with additional regulatory elements such as 35S:BtKAI2 and/or pUBQ:BtKAI2, might be more revealing. As mentioned above, the conclusion that BtKAI2c is not functional requires additional data. Please refer to Burger et al 2019, where transgenic Arabidopsis expressing PpKAI2 homologs show no complementation but were able to bind karrikin KAR1 in vitro.

New data have been added as detailed above in point 2c.

4. Figure 2a and S2a: generation of tree from highly similar sequences introduces uncertainty into the validity of the conclusion made about BtKAI2 ancestry. Also, underrepresentation of plant diversity further weakens the author's conclusion (in line 100-102). If any ancestry can be inferred, the length of branches indicates KAI2b is more similar to the outgroup, thus the conclusion that KAI2a is more ancestral is unsupported. The phylogenetic tree is in agreement with BtKAI2a's similarity to AtKAI2 and its splitting from BtKAI2b and BtKAI2c. Broadening the survey of sequences could provide better support for claims about ancestry.

This is an expansion of point 2b above, which we have addressed by changing our interpretation of the tree. We have not added any new sequences to the tree (there are only so many available within the Brassicaceae) but the new wider analysis in Figure 5 should aid the interpretation about ancestral state of KAI2, at least with respect to the important amino acid residues we have identified here.

5. While the complementation experiments in plants drive the main conclusion of this study, Figure 4 doesn't provide compelling evidence. The use of GR24ent-5DS, a non-natural and unrelated ligand for KAI2, may belie the importance of the residues in conferring karrikin selectivity to BtKAI2s. Given that KAI2s respond poorly to GR24 (or slightly respond to very high concentrations of GR24, as in Figure S9b), the overall effect and statistics (Figure 4a, f) are very mild and perhaps even insignificant if all tests would be scaled to one another. I am wondering if the authors will be able to test the three homologs and the swapped mutants in alternative assays such as ITC or Microdialysis (recently shown with karrikin in Xu et al 2016 and Burger et al 2019). Moreover, the structural modelling (Figure 4b-e) provides little information beyond speculation. The predicted changes in the pocket volume might indeed be attributed by the mild changes in amino acids, yet without a crystal structure (at least for BtKAI2b) or better binding assay, these predictions are very much suggestive. I do think it is important to help readers understand the rational and the exact positions of the residues by 3D models, perhaps as supplementary data and not necessarily as main Figure. In that regard, Lopez-Obando et al 2016, provided similar structural prediction approach for PpKAI2's pockets that proven to be only partially true (Burger et al 2019).

There are many points to address here. We fully recognise the limitations of using DSF and GR24 as a proxy for karrikins in vitro. We used this as a way to test whether the identified amino acids might have an effect on KAI2 function before going to the trouble of making transgenic lines, which are the only true way of really testing the function of the receptors, given the problems with using karrikins in vitro (as mentioned in Discussion). We have our doubts about the validity of conclusions drawn from in vitro assays – for example, what should we conclude if ITC or microdialysis conflicted with the clear plant data? In any case, we attempted ITC and ran into unsurmountable technical issues – specifically, the addition of karrikins caused our protein to precipitate, releasing a very large (and irrelevant) heat signal. We also attempted crystallisation of BtKAI2b but to date have failed to recover any crystals. Given the timeframe for this manuscript (it's been in the making for five years!), we felt it was necessary to move on without additional in vitro data.

As for the validity of the DSF data we present, I am not sure what the reviewer means by “perhaps insignificant if all tests could be scaled to one another”. The effects may be small, but they are robust and they make sense: changes in DSF correspond to changes in planta. It may not be a direct comparison because of the different ligand, but we consider it in the first place as a relatively speedy assay for probing the effects of specific mutations.

We have moved the structural homology models to Supplemental Figure 8, where they are larger and less likely to be mistaken for “real” structural data.

6. It would be helpful to provide a broader in silico analysis in multiple species. This can

strengthen the correlation of the two amino acid residues identified by the authors to KAR1 or KAR2 ligand preference.

Thank you for this suggestion – we agree and this was a very powerful approach given the recent release of the 1KP data. The results are now presented in Fig 5 and Supplemental Table 3, and show that the two amino acids are likely co-dependent.

Minor points:

1. Line 84: missing reference to justify the choice of target genes.

These have now been included, as well as a brief description of how they were identified in *B. tournefortii* (Genbank sequence IDs included at end of manuscript)

2. Detailed quantification of rescue experiments (whole seedling and leaf phenotype) is missing for Figure 2c-d. Full complementation is not supported by the data presented (Line 250), particularly those from germination assays. It would also be helpful if the authors reconciled the differences observed in degree of rescue presented in Figures 3a-b and also provided further statistical tests to ascertain the extent of rescue compared to Ler or *kai2-2*.

We have not performed any further phenotypic characterisation/quantification beyond what is presented. We agree that the germination assays do not fulfil the requirement for full complementation, so we have removed any reference to full complementation and have introduced a new section where we discuss the fact that BtKAI2a activity in seeds is different from that in seedlings (see also response to Reviewer #2). We believe the data in Figure 3 are adequate to show that the seedling phenotype, at least with respect to hypocotyl length, is completely restored in BtKAI2a and b, but not c.

3. In order to better visualize and assess protein molecular weight and levels of expression, authors should provide uncropped gels as supplementary figures. In Figure 2e, the appearance of anti-actin blot and the wide size separation (between 30kD to 60kD on 12% PAGE) should be clarified.

These uncropped images have now been provided in Supplemental Figure 14, as well as an explanation for why we used both KAI2 and GFP antibodies to detect the same proteins. Actin was used as a loading control to verify that equal amounts protein had been loaded in all lanes – this is now specified in the legend to Figure 2.

4. I am wondering if it would be more comprehensive to include BtKAI2c in the differential expression experiment (Figure 3c) as was performed for Figure 3a-b.

We did not include BtKAI2c in gene expression experiments because it greatly increases the workload and expense of the experiment, when we are now very confident that there would be no response in BtKAI2c lines.

5. DSF analyses were performed with His-SUMO tagged KAI2, this is probably related to the solubility state of un-tagged KAI2, but if so, why not testing SUMO-BtKAI2c. Some

clarifications will help to better understand the usage of tagged protein in thermal shift assays.

This has now been expanded upon in the results (lines 217-220) and we included a DSF experiment with cleaved BtKAI2b to show that the presence of the SUMO tag does not dramatically influence the ligand response (Supp Fig 9b). We explain clearly, in a new section of the results, what happens when we try to express BtKAI2c, and the effect of mutations to stabilise the protein.

6. In Figure 4b the label of V191 should be adjusted.

This has been fixed.

7. Line 323: please provide details on promoter used for transient expression besides the name of the vector.

This has now been included, as well as a brief explanation in the results section because we now tried expressing in tobacco using two different sets of plasmids to address Reviewer #2's first concern.

Reviewer #2

Major remarks:

Line 122 and Supplementary figure 5c: Arabidopsis was transformed with GFP tagged BtKAI2s while the expression in tobacco was tested with myc-tagged protein. From the text it reads like the same constructs were tested in both systems. It would be more convincing to show the expression of GFP tagged proteins in tobacco. Other option is to demonstrate that there is no expression the myc-tagged BtKAI2c in Arabidopsis and no complementation of kai2-2 mutant. There is quite some difference in the size of both tags so maybe this could influence the stability of BtKAI2c.

We have included additional experiments using the GFP-tagged proteins in tobacco for BtKAI2c and its mutant variants. These data are now presented in a new Supplementary Figure 6. We found that R98 was fully responsible for the instability of BtKAI2c (and that V96R also disrupts AtKAI2).

Line 129: BtKAI2a and BtKAI2b indeed show differential ligand specificity but only in seedlings. During seed germination BtKAI2a seems to be equally responsive to KAR1 and KAR2. The effect in seeds and seedlings should be clearly separated in the manuscript, because now it seems like BtKAI2a is always more responsive to KAR2, which is not the case. Also, the DLK2 expression was tested in seedlings, which is in agreement with the hypocotyl data, but does not help to understand the specificity, or its lack, of KAI2A during seed germination. I would suggest to check the DLK2 expression in the seeds of BtKAI2a and BtKAI2b lines in response to KAR1 and KAR2 treatment.

This is a fair criticism and we performed the suggested experiment, looking at transcript levels in KAR-treated seeds. DLK2 is very weakly expressed in the mock-treated condition,

making it hard to quantify reliably. Instead we showed KUF1 transcripts. The data support the germination data, which shows that BtKAI2a has no particular preference in seeds, a result that is surprising and interesting (see next comment).

Line 189: It should be specified that this effect is observed only in seedlings. Additional germination assay would be strongly recommended to check if the KAR1 specificity of BfKAI2b can be reversed also in seeds of BtKAI2bV98;V191.

We have done this experiment (Supplemental Fig. 13) and found that BtKAI2bV98;V191 led to a loss of KAR1 preference and, like native BtKAI2a, lost any preference for KAR1 or KAR2 altogether. This supports the data presented in Fig 3a and 3b. There are many possible explanations for this phenomenon but we have proposed improper expression of the transgene in seeds (lines 184-187). We have therefore been careful to stress that ligand preference of BtKAI2a (or proteins with V98/V191) is limited to seedlings.

Supplementary figure 5: - The qPCR analysis should be done with three biological repeats not only one.

The purpose of this experiment was merely to demonstrate that the transgenes were expressed in the BtKAI2c lines, not to provide precise quantification relative to other lines. Rather than perform an expensive and not very informative experiment in triplicate, given the new data explaining the lack of function for BtKAI2c, we decided to remove this figure altogether. Supplemental Figure 5 also shows that GFP-BtKAI2c is faithfully expressed in the transgenic lines.

Minor remarks:

Line 66: The term “KL” should refer to KAI2 ligand not karrikin-like ligand, as there is no evidence, to my knowledge, that this unknown molecule is a karrikin-like.

This is very true, and we have changed it to “KAI2 ligand”.

Line 74: Using the term “KAI2–D14 family” in this sentence but also in throughout the text is confusing. First of all, the manuscript focuses on the KAI2 homologues and their ligand specificity not much is studied for D14. Secondly, DLK2 also belongs to this family but does not respond to strigolactone or karrikins.

We have removed all references to KAI2-D14 family. We used it as a shorthand for butenolide receptors that this family comprises, but we accept that DLK2 is a member and the functions of this family may be very diverse.

Lines 83-86: There is no difference in the expression of both marker genes at 100 nM so stating that “ten-fold higher concentrations of KAR2 (1 μ M) compared with KAR1 (100 nM) was required to reach equivalent levels” is a bit unclear. I would rather describe the significant difference in marker gene expression that is clear when 1 μ M concentration is used.

We have adjusted this sentence to clarify the point.

Line 91: I suggest to rewrite the sentence as it sounds a bit strange that the expression was more responsive... The genes were more responsive or the expression was changed?

We agree, and this sentence has been improved: "...were significantly more highly expressed when treated with 1 μ M KAR₁ than with 1 μ M KAR₂".

Line 141: This sentence is a bit overstatement. BtKAI2a is also responsive to KAR1 so BtKAI2b contributes to the enhanced KAR1-responsiveness in seeds rather than is fully responsible for it.

Agreed, we have changed the sentence to read "As *BtKAI2b* is more highly expressed than *BtKAI2a* in *B. tournefortii* seeds and seedlings (Fig. 2b), we conclude that the ligand specificity of BtKAI2b substantially contributes to the enhanced KAR₁-responsiveness of this species at both of these stages of the life cycle."

Line 181: It should be mentioned that in this case also KAI2 promoter was used.

Thank you, we have now included this information.

Lines 293-295: It should be indicated how the treatment with karrikins was done in *Brassica tournefortii* seeds.

This information has now been explicitly stated: "*Brassica tournefortii* seeds were imbibed on glass fibre filters in the dark at 22 °C and treated with karrikins using the same procedure as described under seed germination assays, above." – and in the seed germination assays section, it now reads "...held in 9-cm petri dishes and supplemented with mock or karrikin treatments (3 mL of aqueous treatment solution per petri dish)."

Figure 1b and Supplementary figure 1 and 7, line 375, 377, 436: the word "seed" should be in a plural form in several places. I suggest to check it carefully in the whole manuscript.

Technically you can use "seed" as a collective singular to refer to a batch of seeds as a unit (for a nice explanation from 1987, see <http://wssa.net/wp-content/uploads/Seed-Seeds-and-Seedlings.pdf>). But we agree for the benefit of all readers that one should be consistent, and have corrected to "seeds" throughout.

Figure 1c,d: Hypocotyl elongation responses of *B. tournefortii* seedlings treated with KAR1 and KAR2 and grown for four days under continuous red light.- I would rewrite the sentence to make clear that seedlings were grown for 4 days on the medium with KAR1 or KAR2.

Thank you, we have clarified this in the figure legend.

Figure 2: Instead of "treated with or without 1 μ M KAR" I would rather write treated mock or with 1 μ M KAR.

Agreed - this now reads "treated with 0.1% acetone or with 1 μ M KAR₁ for 24 h."

Supplementary figure 2 and 3: The reference in figure legend should be changed to a number to fit a reference style of the manuscript.

These have been fixed, but because they are supplementary figures, the numbers correspond to different references from those in the main text.

Supplementary figure 5: Consider rewriting the sentence: "RNA was isolated from approximately 50 seven-day-old seedlings per genotype, and are the same samples as those shown in the immunoblots in Figure 2."

This has been removed along with the corresponding data as mentioned above.

REVIEWERS' COMMENTS:

Reviewer #1 (Remarks to the Author):

The authors made a remarkable effort to address the issues raised by the reviewers. This revised version has improved its impact and clarity. I have no additional major comments.

I found few minor edits throughout the text:

1. Line 194 – consider editing (Nelson 2010) with a correct reference style.
2. Consistency with naming species, for example – the authors sometimes use *Brassica tournefortii* or *B. tournefortii*, I was wondering if the authors can consider picking one and use it consistently throughout the text.
3. 6xHIS-SUMO tag was used in all BtKAI2s experiments, yet in all cases only the SUMO-BtKAI2 is denoted which can be misleading. The authors should consider denoting the His tag as well throughout the text and figures (HIS-SUMO-BtKAI2). Alternatively, the authors can clarify this notion. Perhaps rephrase Lines 220-224 something like: " and therefore we used intact 6xHIS-SUMO fusion proteins (denoted SUMO-BtKAI2s throughout...)...."

Reviewer #2 (Remarks to the Author):

Three homologues of KAI2 has been characterized in *B. tournefortii*, of which two show different preferences for KAR1 and KAR2. Two amino acid residues were identified that are responsible for this ligand specificity. To my opinion, the manuscript by Sun et al. has been significantly improved and all the remarks were addressed in the satisfactory manner.

I have only a few minor remarks:

Line 194: the format of the reference should be changed.

Quantification of plant height and branching parameters should be added to methods section.

Line 501: add information on the transformation of d14 mutant as well.

Figure 4 d: the graph legend (0.1 μ M KAR1 and KAR2) does not fit with the figure legend (1 μ M).

Response to reviewers' comments

Reviewer #1 (Remarks to the Author):

The authors made a remarkable effort to address the issues raised by the reviewers. This revised version has improved its impact and clarity. I have no additional major comments.

I found few minor edits throughout the text:

1. Line 194 – consider editing (Nelson 2010) with a correct reference style.

This has been corrected.

2. Consistency with naming species, for example – the authors sometimes use *Brassica tournefortii* or *B. tournefortii*, I was wondering if the authors can consider picking one and use it consistently throughout the text.

Thank you for pointing this out – we have used the full name when first mentioned in the abstract, introduction and first figure, and abbreviated name thereafter.

3. 6xHIS-SUMO tag was used in all BtKAI2s experiments, yet in all cases only the SUMO-BtKAI2 is denoted which can be misleading. The authors should consider denoting the His tag as well throughout the text and figures (HIS-SUMO-BtKAI2). Alternatively, the authors can clarify this notion. Perhaps rephrase Lines 220-224 something like: “ and therefore we used intact 6xHIS-SUMO fusion proteins (denoted SUMO-BtKAI2s throughout...)....”

This has been addressed. We state that “SUMO” is shorthand for 6xHIS-SUMO and then use the “SUMO-“ prefix for all appropriate places in the text and in the figures.

Reviewer #2 (Remarks to the Author):

Three homologues of KAI2 has been characterized in *B. tournefortii*, of which two show different preferences for KAR1 and KAR2. Two amino acid residues were identified that are responsible for this ligand specificity. To my opinion, the manuscript by Sun et al. has been significantly improved and all the remarks were addressed in the satisfactory manner.

I have only a few minor remarks:

Line 194: the format of the reference should be changed.

Please see above

Quantification of plant height and branching parameters should be added to methods section.

This has now been included in the “Plant material and measurement of growth” section of the methods.

Line 501: add information on the transformation of d14 mutant as well.
Figure 4 d: the graph legend (0.1 μM KAR1 and KAR2) does not fit with the figure legend (1 μM).

Thank you for pointing this out; we erroneously copied the wrong graph legend. The figure legend text is correct. The figure itself has been corrected.